# Effect of blockage on wind turbine power and wake development

Olivier Ndindayino , Augustin Puel , and Johan Meyers

Department of Mechanical Engineering, KU Leuven, Celestijnenlaan 300, B3001 Leuven, Belgium

**Correspondence:** Olivier Ndindayino (olivier.ndindayino@kuleuven.be)

**Abstract.** Recent work by Lanzilao & Meyers (J. Fluid Mech, 2024) has shown that wind-farm blockage introduces an unfavourable pressure gradient in front of the farm and a favourable pressure gradient in the farm, which are strongly correlated with the nonlocal efficiency and wake efficiency respectively. In particular, the favourable pressure gradient in the farm increases the farm wake efficiency, defined as the average farm power normalized by the average front-row power. Here, we

investigate the impact of blockage on wake development and power of wind turbines using an idealized large-eddy simulation setup in which blockage conditions are artificially introduced using a rigid-lid, further also using neutral stratification and no wind veer. We simulate both infinite and finite single turbine rows, as well as a setup with two staggered rows. Blockage strength is adjusted by varying the boundary layer height ($H$) and turbine spacing ($S$). We find that blockage strongly affects near wake behaviour, altering Froude momentum theory, by introducing a favourable pressure difference ($\Delta p_{NW}$) across the

turbine row. The same setup also leads to an unfavourable pressure difference ($\Delta p_{FW}$) in the far wake, which simply follows from the rigid-lid conditions and the change of momentum flux due to wake recovery. A strong positive correlation was observed of $-\Delta p_{NW}$ with both power coefficient ($C_P$) and thrust coefficient ($C_T$). Specifically, as $S$ and $H$ decrease, $-\Delta p_{NW}$, $C_P$ and $C_T$ increase. At the same time a lower induction is observed at the rotor disk, and a lower wake deficit in the near wake. The reduction of near wake velocity deficit as a result of blockage also translates into lower deficits and wake widths in

the far wake. When scaling the far wake development with initial far wake deficit and width, we do not see a direct effect of the adverse pressure gradient on the wake recovery. However, we do see a profound effect of $H$ on the wake spreading, with higher boundary layers leading to faster spreading. This relates to the fact that, the wake can more freely expand vertically in high-boundary layer cases, into a larger region of high-speed flow than for shallow boundary layers. Finally, we introduce a simplified Froude momentum balance to parametrize the relation between blockage, pressure drop and near wake properties,

and compare it to the LES results.

## 1 Introduction

In large wind farms, significant flow slowdown occurs upstream of the farm, called blockage. Measurements conducted by (Bleeg et al., 2018) before and after wind farm commissioning revealed significant reductions of wind speed upstream of the farm (up to 4% for 10 rotor diameters). Around the same time Allaerts and Meyers (2017, 2018) described the excitation of

25 gravity waves in their wind-farm large-eddy simulations, also observing significant slow down of wind speed in front of the farm. Since then, various other studies have also discussed the excitation of gravity waves by wind farms, and related blockage

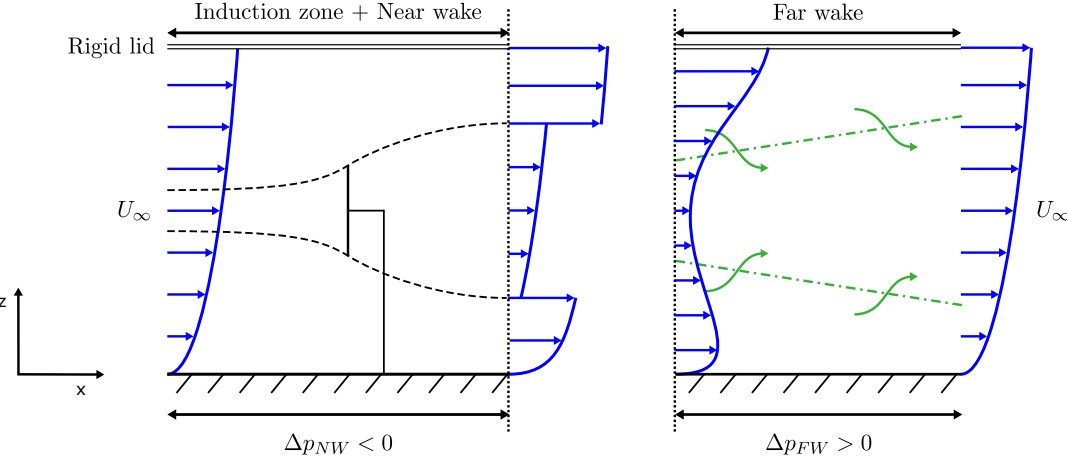

**Figure 1.** Sketch (side view) of a single turbine row (infinitely wide) inside an idealised ABL with rigid-lid top condition.

effects (Wu and Porté-Agel, 2017; Maas, 2023; Stipa et al., 2024). Recently, Lanzilao and Meyers (2024) performed LES simulations of a fixed 1.6 GW wind farm in different atmospheric stratified conditions to investigate effects on wind-farm efficiency, blockage and related gravity-wave excitation. They found that the upstream slowdown associated with blockage is
dominantly originating from the vertical displacement of the capping inversion at the top the boundary layer, and the associated hydrostatic pressure induced by the increased column of cold (higher density) air below the inversion. For the atmosphere conditions considered in their simulations, purely hydrodynamic blockage effects (i.e. associated with Bernoulli's law and induction by the turbines, see also Segalini and Dahlberg, 2020) were at least an order of magnitude smaller (Lanzilao and Meyers, 2022, 2024).

Similar to Allaerts and Meyers (2017, 2018), Lanzilao and Meyers (2024) identified a strong unfavourable pressure gradient upstream of the farm, but also a favourable pressure gradient within the wind farm. They defined the non-local efficiency $\eta_{nl} = P_1/P_\infty$ as the ratio between the power of a free standing turbine $P_\infty$ and the average turbine power of the first row of a wind farm $P_1$, and a wake efficiency $\eta_w = P_{tot}/(N_t P_1)$ as the ratio between the average turbine power in the wind farm $(P_{tot}/N_t)$ and the average turbine power of the first row. A strong negative correlation was found between the unfavourable
upstream pressure gradient and $\eta_{nl}$, and a strong positive correlation between the favourable pressure gradient within the farm and $\eta_w$. It was further found that, depending on atmospheric conditions, the beneficial effects of the favourable pressure gradient can offset the negative effects of the upstream unfavourable pressure gradient, sometimes leading a larger farm efficiency than in a similar fully neutral case without free-atmosphere stratification.

    In the current manuscript, our aim is to better understand the relationship between favourable pressure gradient and improved
efficiency using a new set of large-eddy simulations. To this end, we strongly simplify the simulation setup by replacing blockage induced by free-atmosphere stratification with blockage induced by a rigid-lid at the top of the boundary layer. Such a rigid-lid condition could be the equivalent of an infinitely strong capping inversion. Although such a condition cannot exist, since the strongest density jump conceivable over the capping inversion should be well below the difference between

density at 1 atm and vacuum, rigid-lid blockage nonetheless shows some similarities with blockage induced by free-atmosphere stratification (Smith, 2024). The advantage of using a rigid-lid condition is that the relationship between blockage, pressure gradient and efficiency is easier to quantify. This is illustrated in Fig. 1, for a single infinitely wide row of turbines. Looking at a control volume around the near wake and induction region of a turbine, it is directly clear that a favourable pressure difference should exist ($\Delta p_{NW} < 0$) – presuming negligible friction at the ground, this is a direct result from Newton's second law, the presence of the turbine thrust force, and continuity (so that outflowing momentum is larger than inflowing momentum). Similarly, an unfavourable pressure difference ($\Delta p_{FW} > 0$) should apply in the far wake. Moreover, applying the conservation of momentum in the streamwise direction, considering the entire domain shown in Fig. 1 for an infinite row; that is from the beginning of the induction zone to the end of the far wake, as wide as the turbine spacing $S$ and as high as the boundary layer height $H$, we find

$$HS(\Delta p_{NW} + \Delta p_{FW} + \Delta p_{bg}) + F_T + F_{fric} = 0, \tag{1}$$

where $F_T$ ($> 0$) is the turbine thrust, $F_{fric}$ ($> 0$) is the friction from the ground and $\Delta p_{bg}$ ($< 0$) is the background pressure difference in the absence of the turbines. We remark that the velocity profile at the end of the far wake has recovered back to the velocity profile at the start of the induction zone, therefore no change in momentum is present in Eq. (1). We assume that $F_{fric} \approx \Delta p_{bg}$, thus $\Delta p_{NW}$ and $\Delta p_{FW}$ are dynamic pressure perturbations, superimposed on the background pressure. This simplifies Eq. 1 to $-\Delta p_{NW} \approx \Delta p_{FW} + F_T/HS$, and thus $-\Delta p_{NW} > \Delta p_{FW}$, from which we hypothesize that the effect of blockage on efficiency is mostly related to changes in the near-waking behaviour.

The effects of domain blockage by rigid boundaries on turbine performance have been studied before, mostly in the context of blockage corrections in wind tunnel experiments of single model wind turbines (Mikkelsen and Sørensen, 2002; Werle, 2010; Segalini and Inghels, 2014). In these studies, the focus was on estimating free-flow conditions, thus eliminating the effects of wind tunnel walls. These corrections, were built by extending Froude momentum theory (Froude, 1889) to account for the pressure gradients included by any (small) domain blockage present in the wind tunnel. Here, we will use similar ideas to develop a simple model that correlates favourable pressure gradients induced by blockage with near wake induction and turbine power extraction, but now to quantify the effect of blockage rather than to exclude it. A similar approach was already used by Garrett and Cummins (2007) for tidal turbines in constant water level channels, and some elements were also used by Nishino and Willden (2012, 2013) for modelling the blockage effect of a finite array of tidal turbines in a wide channel cross-section.

The setup of a row of closely spaced turbines that we are studying in the current work has also been explored in the past by McTavish et al. (2013, 2015); Strickland and Stevens (2020, 2022). They reported so called 'in-field' blockage, leading to increased power production of the turbines, which they attributed to mutual favourable interactions between turbines. However, by performing a careful domain sensitivity study, Bleeg and Montavon (2022) later showed that these beneficial effects disappear when the domain size is sufficiently large. Thus, in these earlier reports, power increase due to 'in-field' blockage was effectively a result of domain blockage, and consequently may be related to the increased wake efficiency observed by Lanzilao and Meyers (2024) under favourable pressure gradient conditions.

Looking at the sketch in Fig. 1 and referring to the discussion above, an unfavourable pressure gradient is expected in the far wake, albeit smaller than the favourable pressure gradient in the near wake. The effects of pressure gradients on wake development have been studied by Liu et al. (2002); Shamsoddin and Porté-Agel (2017) for plane wakes, and by Shamsoddin and Porté-Agel (2018) for axisymmetric wakes, and later more general conditions were also considered by Dar and Porté-Agel (2022). All of these studies conclude that unfavourable pressure gradients lead to slower, and favourable pressure gradients to faster wake recovery. However, we note already that for all simulations considered in the current manuscript, we found the unfavourable pressure gradients in the far wake to be much smaller than the values appearing in above studies, so that wake recovery is unaffected by them.

In the current work, we setup a range of large-eddy simulations of a neutral rigid-lid pressure driven boundary layer, in which we represent both infinite and finite single turbine rows, as well as a setup with staggered rows. Blockage strength is adjusted by varying the boundary layer height ($H$) and turbine spacing ($S$). Simulation results are further compared to a simple model that extends classical Froude momentum theory, parametrizing the effects of pressure gradients on wind turbine power, thrust, and induction. Such a model can straightforwardly be incorporated into engineering blockage and wake models such as, e.g., WAYVE Allaerts and Meyers, 2019; Devesse et al., 2022; Stipa et al., 2024; Devesse et al., 2024a, b, in which large-scale pressure gradients coming from free-atmosphere stratification and gravity wave feedback are available. The latter is however a topic of future research and not in the scope of the current manuscript.

The article is structured as follows. The setup of the LES simulations are elaborated in Sect. 2. The extension of the Froude momentum theory is discussed in Sect. 3. Next, Sect. 4 presents the LES results, for the far and near wake analysis, including model validation. Lastly, conclusions are provided in Sect. 5.

## 2 Methodology

### 2.1 Governing equations

We consider the filtered Navier–Stokes equations for a neutral pressure-driven boundary layer, given by

$$\frac{\partial \tilde{u}_i}{\partial x_i} = 0, \tag{2}$$

$$\frac{\partial \tilde{u}_i}{\partial t} + \tilde{u}_j \frac{\partial \tilde{u}_i}{\partial x_j} = -\frac{1}{\rho}\frac{\partial \tilde{p}^*}{\partial x_i} - \frac{1}{\rho}\frac{\mathrm{d}p_\infty}{\mathrm{d}x_1}\delta_{1i} - \frac{\partial \tau_{ij}^{sgs}}{\partial x_j} + f_i, \tag{3}$$

where, the horizontal and vertical directions are represented by indices $i = 1, 2$ and 3, with $(x_1, x_2, x_3) = (x, y, z)$. The filtered velocity components are denoted by $\tilde{u}_i$ for the three-dimensional flow field, with $(\tilde{u}_1, \tilde{u}_2, \tilde{u}_3) = (\tilde{u}, \tilde{v}, \tilde{w})$. The filtered modified pressure is defined as $\tilde{p}^* = \tilde{p} - p_\infty + \rho_0 \tau_{kk}/3$, where $p_\infty$ represents the mean background pressure, and $\tau_{kk}/3$ denotes the trace of the subgrid-scale stress tensor $\tau_{ij} = \widetilde{u_i u_j} - \tilde{u}_i \tilde{u}_j$, and where a subgrid-scale model is used to model the anisotropic component of the residual-stress tensor $\tau_{ij}^{sgs} = \tau_{ij} - \delta_{ij}\tau_{kk}/3$. Note that the direct effect of viscosity on resolved scales in the LES is negligible; all the dissipation is handled by the subgrid-scale stresses. The forces (per unit of density) $f_i$ exerted by

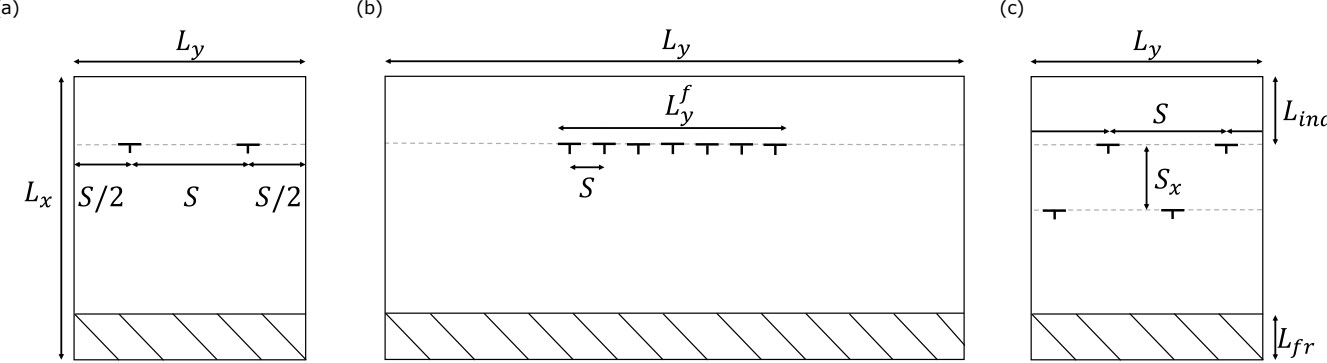

**Figure 2.** The plane view of (a) infinite row, (b) finite row and (c) infinite staggered rows computational main domain.

the wind turbines on the flow are modelled using a non-rotating actuator disk model (Goit and Meyers, 2015) with a Shapiro correction factor (Shapiro et al., 2019) to avoid over prediction of turbine power on typical LES grid resolutions.

The governing equations (2–3) are solved using the SP-Wind solver, an in-house software developed at KU Leuven (Calaf et al., 2010; Goit and Meyers, 2015; Munters et al., 2016b; Allaerts and Meyers, 2017; Lanzilao and Meyers, 2024). The equations are integrated over time using a standard fourth-order Runge–Kutta scheme, with the time step determined by a Courant–Friedrichs–Lewy (CFL) number of 0.4. Discretization in horizontal directions is performed using a Fourier pseudo-spectral method, employing the 3/2 dealising rule. In the vertical direction, an energy-preserving fourth-order finite difference

scheme is applied (Verstappen and Veldman, 2003). Continuity is enforced by a direct solve of the Poisson equation at each stage of the Runge–Kutta method. The influence of subgrid-scale motions on the resolved flow is captured by the Smagorinsky model in combination with Mason and Thomson (1992) wall damping, using a Smagorinsky length $\ell_s^{-n} = (C_s\Delta)^{-n} + [\kappa(z + z_0)]^{-n}$, with $C_s = 0.14$, and $n = 1$, $\Delta = (\Delta x \Delta y \Delta z)^{1/3}$ the grid spacing, and $\kappa = 0.41$ the Von Karman constant. This is consistent with prior studies using SP-Wind (Meyers, 2011; Allaerts and Meyers, 2017; Lanzilao and Meyers, 2024). We also

refer to Calaf et al. (2010); Lignarolo et al. (2016); Martínez-Tossas et al. (2018); Sood et al. (2022) for code benchmarking, and validation.

### 2.2   Simulation setup

Figure 2 provides an overview of the different simulations setups considered in the current study, and a list of all cases is provided in Table 1. All turbines correspond to the International Energy Agency (IEA) 15 MW offshore reference wind turbine

(Gaertner et al., 2020), with hub height $z_h = 150$ m and diameter $D = 240$ m. Conditions are such that the turbines operate below their rated power. Consequently, a constant disk-based thrust coefficient is $C_t' = 1.44$ (Calaf et al., 2010), is used as an input parameter for the simulations.

First of all, a set of single infinite-row simulations are considered (Fig. 2a). We select a domain width $L_y = 9.6$ km with spanwise periodic boundary conditions, and by varying the number of turbines in the domain, we can change the turbine

spacing $S$. Secondly, we also consider a finite row setup consisting of 7 turbines (Fig. 2b). Here, we take $L_y = 50$ km, so

**Table 1.** Overview of the wind farm simulation cases and their computational domain dimensions $L_x \times L_y \times H$, the turbine spacing $S/D$ and row averaged power coefficient $\langle C_P \rangle_{row}$ and thrust coefficient $\langle C_T \rangle_{row}$ (values of both rows are shown for the infinite staggered row simulations).

| Cases | $L_x \times L_y \times H$ (km$^3$) | $S/D$ | $\langle C_P \rangle_{row}$ | $\langle C_T \rangle_{row}$ |
|---|---|---|---|---|
| Inf-H350-S2.5 | $20 \times 9.6 \times 0.35$ | 2.5 | 0.6993 | 0.8894 |
| Inf-H350-S5 | $20 \times 9.6 \times 0.35$ | 5 | 0.6358 | 0.8347 |
| Inf-H350-S10 | $20 \times 9.6 \times 0.35$ | 10 | 0.6031 | 0.8058 |
| Inf-H350-S20 | $20 \times 9.6 \times 0.35$ | 20 | 0.5895 | 0.7936 |
| Inf-H350-S40 | $20 \times 9.6 \times 0.35$ | 40 | 0.5818 | 0.7866 |
| | | | | |
| Inf-H500-S2.5 | $20 \times 9.6 \times 0.50$ | 2.5 | 0.6639 | 0.8588 |
| Inf-H500-S5 | $20 \times 9.6 \times 0.50$ | 5 | 0.6172 | 0.8181 |
| Inf-H500-S10 | $20 \times 9.6 \times 0.50$ | 10 | 0.5969 | 0.7998 |
| Inf-H500-S20 | $20 \times 9.6 \times 0.50$ | 20 | 0.5842 | 0.7887 |
| Inf-H500-S40 | $20 \times 9.6 \times 0.50$ | 40 | 0.5802 | 0.7849 |
| | | | | |
| Inf-H700-S2.5 | $20 \times 9.6 \times 0.70$ | 2.5 | 0.6386 | 0.8366 |
| Inf-H700-S5 | $20 \times 9.6 \times 0.70$ | 5 | 0.6051 | 0.8069 |
| Inf-H700-S40 | $20 \times 9.6 \times 0.70$ | 40 | 0.5757 | 0.7809 |
| | | | | |
| Fin-H500-S2.5 | $20 \times 50 \times 0.50$ | 2.5 | 0.6113 | 0.8128 |
| Fin-H500-S5 | $20 \times 50 \times 0.50$ | 5 | 0.6029 | 0.8054 |
| | | | | |
| Inf-H500-S5-stag | $20 \times 9.6 \times 0.50$ | 5 | 0.6169; 0.6156 | 0.8177; 0.8166 |
| Inf-H500-S10-stag | $20 \times 9.6 \times 0.50$ | 10 | 0.5922; 0.5939 | 0.7965; 0.7982 |

that $L_y/L_y^f \approx 7$ for the case with widest spacing ($S = 5$), following Lanzilao and Meyers (2024) who recommend $L_y/L_y^f \geq 6$ to avoid artificial effects from spanwise boundaries. Finally (Fig. 2c), two infinitely wide staggered rows of turbines are also considered, with $S_x/D = 5$, also using a domain width of $L_y = 9.6$ km.

The simulation domain $L_x \times L_y \times H$ is further selected as follows. All simulations use $L_x = 20$km, with an upstream region $L_{ind} = 4$ km in front of the turbines, and a fringe region length $L_{fr} = 2.6$ km. Different boundary layer heights are considered, i.e. $H = 350$, 500 and 700 m. All simulations use a wall stress boundary conditions at the bottom (with a surface roughness $z_0 = 10^{-4}$ m, which is a typical offshore value (Taylor and Yelland, 2001)), and symmetry conditions at the top. Boundary conditions in the horizontal directions are periodic, as a result of our pseudo-spectral discretization method. To break the periodicity in the streamwise direction and prescribe an inflow condition, we use a fringe-region technique to drive the main domain by turbulent fully developed statistically steady flow fields obtained from a concurrent precursor simulation (Stevens

et al., 2014). Using a precursor simulation ensures that turbulent inflow conditions are generated directly by the Navier-Stokes equations, thereby enhancing the accuracy and realism of the inflow turbulence (Munters et al., 2016a).

To prevent persistent spanwise locking of large-scale streamwise turbulent structures, a shifted periodic boundary condition is applied within the fringe-region, bypassing the need of excessive streamwise domain lengths (Munters et al., 2016b).

The precursor simulation domain $L_x^p \times L_y^p \times H$ is defined as follows. Three different precursor simulations are performed for the infinite row cases, i.e. $L_x^p = 10$ km and $L_y^p = 9.6$ with a height of $H = 350, 500$ and $700$ m. One additional precursor simulation is performed for the finite row cases, with $L_x^p = L_y^p = 10$ km and $H = 500$ m. However, since the fringe region spans the full width and height of the main domain, SP-Wind requires matching heights and widths between the precursor and main domain when they run concurrently. To achieve this, the tiling technique of Sanchez Gomez et al. (2023) is used to extend
the precursor flow fields in the y direction from 10 to 50 km.

All precursor simulations are driven by a constant pressure gradient $\mathrm{d}p_\infty/\mathrm{d}x_1 = -u_\tau^2/H$, with a friction velocity $u_\tau \approx 0.275$ ms$^{-1}$ typical for offshore conditions (Lanzilao and Meyers, 2024). Fixing $u_\tau$ yields approximately the same hub-height wind speed for a given $z_0$ and varying $H$ simulations. Finally, all simulations and domains use the same grid resolution, i.e. $\Delta x = 40$ m, $\Delta y = 24$ m and $\Delta z = 7.93$ m in the streamwise, spanwise and vertical directions, respectively. This is consistent with
resolutions used earlier in e.g., (Allaerts and Meyers, 2017, 2018; Lanzilao and Meyers, 2024).

Precursor simulations are initialized using a log profile $u = (u_\tau/\kappa)ln(z/z_0)$ in combination with random noise and simulated for 20000 seconds, such that a fully developed statistically stationary pressure-driven boundary layer can develop. Subsequently, the precursor domain and main domain are run concurrently for another 3000 seconds for the main domain to fully develop given the precursor inflow. Finally, precursor and main domain are concurrently progressed for an averaging time
$T_{av}$ of 10000 seconds.

## 3    Froude momentum theory with blockage

We extend the Froude momentum theory to parametrize the relation between blockage, pressure drop and near wake properties for a single turbine present within a row or farm. The model is an extension of classic axial momentum theory developed by Rankine (1865) and Froude (1889). A similar approach was, e.g., used by Werle (2010) and Segalini and Inghels (2014) to
derive blockage corrections for wind tunnels, or by Garrett and Cummins (2007) in the context of tidal turbines, although we start from a slightly more general formulation that does not define inlet and outlet areas of the control volume in advance (see Fig. 3).

Consider a control volume around a single turbine within a row or farm (Fig. 3) that is based on a streamtube, extending from the the start of the induction zone until the end of the near wake. We choose the area $A_1 = SH$, which corresponds to the
'available' inflow for the turbine. Note that the outlet area $A_2$ is a priori unknown, but expected to be larger than $A_1$. In the case of an infinite row of turbines, $A_2 = A_1$ is obtained. Moreover, similarly to Garrett and Cummins (2007) and Werle (2010), $A_1$ and $A_2$ do not need to be cylindrical as the momentum theory is a one dimensional analysis which does not intrinsically define the cross-sectional shapes. We refer to Fig. A1 in appendix A, for a visualization of the control volume around turbines in an

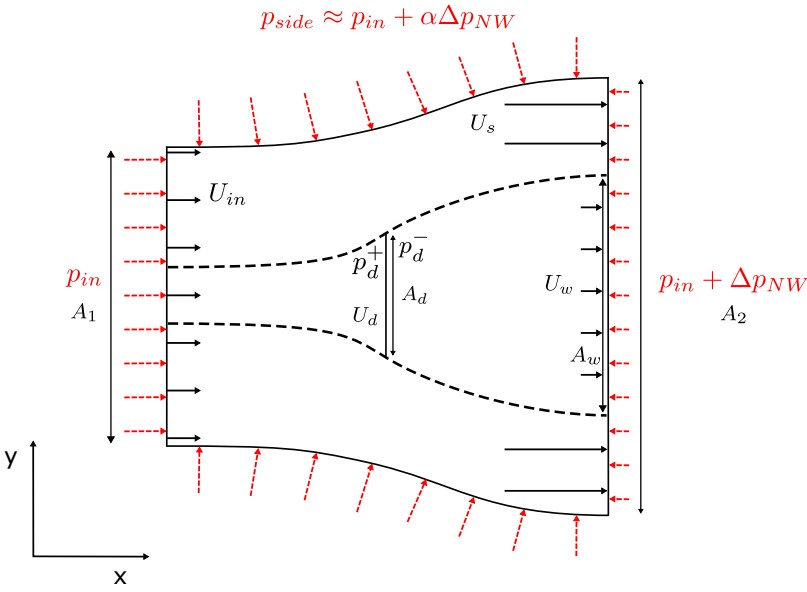

**Figure 3.** Top view of the general setup of the control volume around a turbine to define blockage effects.

infinite and finite row, based on the streamlines calculated from the LES. We further define the disk area $A_d = \pi D^2/4$, and the wake area $A_w$, which is not known a priori.

We presume uniform inflow $U_{in}$ into the control volume, and uniform pressures $p_{in}$ and $p_{in} + \Delta p_{NW}$ over inlet and outlet, respectively. Furthermore, the disk velocity $U_d = U_{in}(1-a)$, with $a$ the axial induction factor. This leads to a thrust force $F_T = 1/2\rho C_T' U_{in}^2 (1-a)^2 A_d$, and in absence of drag or mechanical losses, also to $P = 1/2\rho C_T' U_{in}^3 (1-a)^3 A_d$.

First, applying the principle of conservation of mass on the streamtube that passes through the rotor and the entire streamtube, we obtain

$$A_d U_{in}(1-a) = A_w U_w, \tag{4}$$

$$A_w U_w + (A_2 - A_w)U_s = A_1 U_{in}. \tag{5}$$

Secondly, we apply the principle of conservation of momentum in the streamwise direction over the entire streamtube. Similar to Eq. (1), we presume that the friction at the ground is balanced by the background pressure difference present in the absence of a turbine ($F_{fric} \approx \Delta p_{bg}$), so that, to first order, $\Delta p_{NW}$ contains the pressure perturbations due to the presence of the wind turbines only. A similar strategy was, e.g., used by Kirby et al. (2023). As a result, the momentum balance equation is

$$-\frac{1}{2}C_T' U_{in}^2 (1-a)^2 A_d - \frac{1}{\rho}\alpha \Delta p_{NW}(A_1 + A_2) = (A_2 - A_w)U_s^2 + A_w U_w^2 - A_1 U_{in}^2, \tag{6}$$

where the density $\rho$ is a known constant value. Here, we have further presumed that the average pressure on the mantle of the streamtube, corresponds approximately to $p_{side} \approx p_{in} + \alpha \Delta p_{NW}$. Thirdly, using $F_T = (p^+ - p^-)A_d$, and eliminating $p^+$ and $p^-$ using Bernoulli for streamlines in front and behind the disk respectively (similar to what is done in derivations of the

classical Betz theory), we arrive at

$$\frac{1}{2}C_T'U_{in}^2(1-a)^2 = \frac{1}{2}U_{in}^2 - \frac{1}{2}U_w^2 - \frac{1}{\rho}\Delta p_{NW}. \tag{7}$$

Finally, using Bernoulli for streamlines that do not pass through the rotor area, we also find

$$\frac{1}{2}U_{in}^2 = \frac{1}{\rho}\Delta p_{NW} + \frac{1}{2}U_s^2 \tag{8}$$

Above leads to a set of five model equations: (Eqs. 4–8), with in principle four known input variables $A_1, A_d, C_t'$ and $U_{in}$, which leaves six unknown variables: $a, U_w, U_s, A_w, A_2$, and $\Delta p_{NW}$. Thus we lack one equation to arrive at a closed system.

There are two situations that lead to a simple closure of the system. First of all, imposing $\Delta p_{NW} = 0$ leads to the classical Betz–Joukowsky theory for a single isolated turbine. The other case corresponds to a infinite row of turbines, in which case $A_2 = A_1 = SH$ (See appendix A, Fig. A1a). This leads to the wind tunnel blockage corrections earlier discussed (e.g., Werle,

2010 and Segalini and Inghels, 2014; note that a converging or diverging wind tunnel would require a known $A_2 \neq A_1$).

In the more general setting of a finite row of turbines, neither $A_2$ nor $\Delta p_{NW}$ are known a priori. At the sides of a finite row, turbines have clearly more space to expand sideways, and possibly a lower $\Delta p_{NW}$ applies than in the centre of the row, where the expansion of the inflow area $A_1$ is hindered by the surrounding turbines (See appendix A, Fig. A1a). Thus, in such a system a coupling through a larger pressure system can be expected, with a pressure gradient not only in the streamwise direction,

but also in the spanwise direction. An additional relation for the pressure system, may be, e.g., obtained from an atmospheric perturbation model (Allaerts and Meyers, 2019; Devesse et al., 2022; Stipa et al., 2024; Devesse et al., 2024a, b; see also the open-source model WAYVE) which incidentally also parametrize the more complex relation between capping inversion displacement and gravity wave excitation in wind farms. It is however not in the scope of the current work to develop such a coupling. Instead, we will evaluate the pressure system arising in our large-eddy simulations in more detail, and use $\Delta p_{NW}$

measured from the simulations as an additional input to evaluate the system above, comparing its output in terms of power and thrust versus those obtained directly from LES.

## 4 Results and discussion

### 4.1 Velocity deficit and wake recovery

#### 4.1.1 Induction zone and near wake analysis

Here we look in more detail into the wake development for the different simulation cases. In Fig. 4 we present the streamwise centreline velocity ($U_c$), scaled by the free-stream velocity at hub height ($U_\infty$) and averaged over the turbines in each row, for all cases, providing a general overview of the differences and possible similarities in wake development. In Fig. 4a two regions of interest are depicted, i.e. the induction zone and near wake on the one hand, and the far wake on the other hand. We use the minimum of the velocity to mark the end of the near wake region, which can depend significantly on the boundary layer

height $H$ as, e.g., seen in Fig. 4a. The start of the far wake region is selected $x/D = 6$ for all cases. As we will show below (see

Fig. 6a and b), the far wake velocity profile can be fitted very well with a Gaussian function, which we can use to quantify the wake recovery behaviour in detail. Upstream of our selected starting point for the far wake, such a fit does not work very well, as the wake profile is transitioning from a top-hat to a Gaussian shape, potentially requiring more advanced fitting shapes.

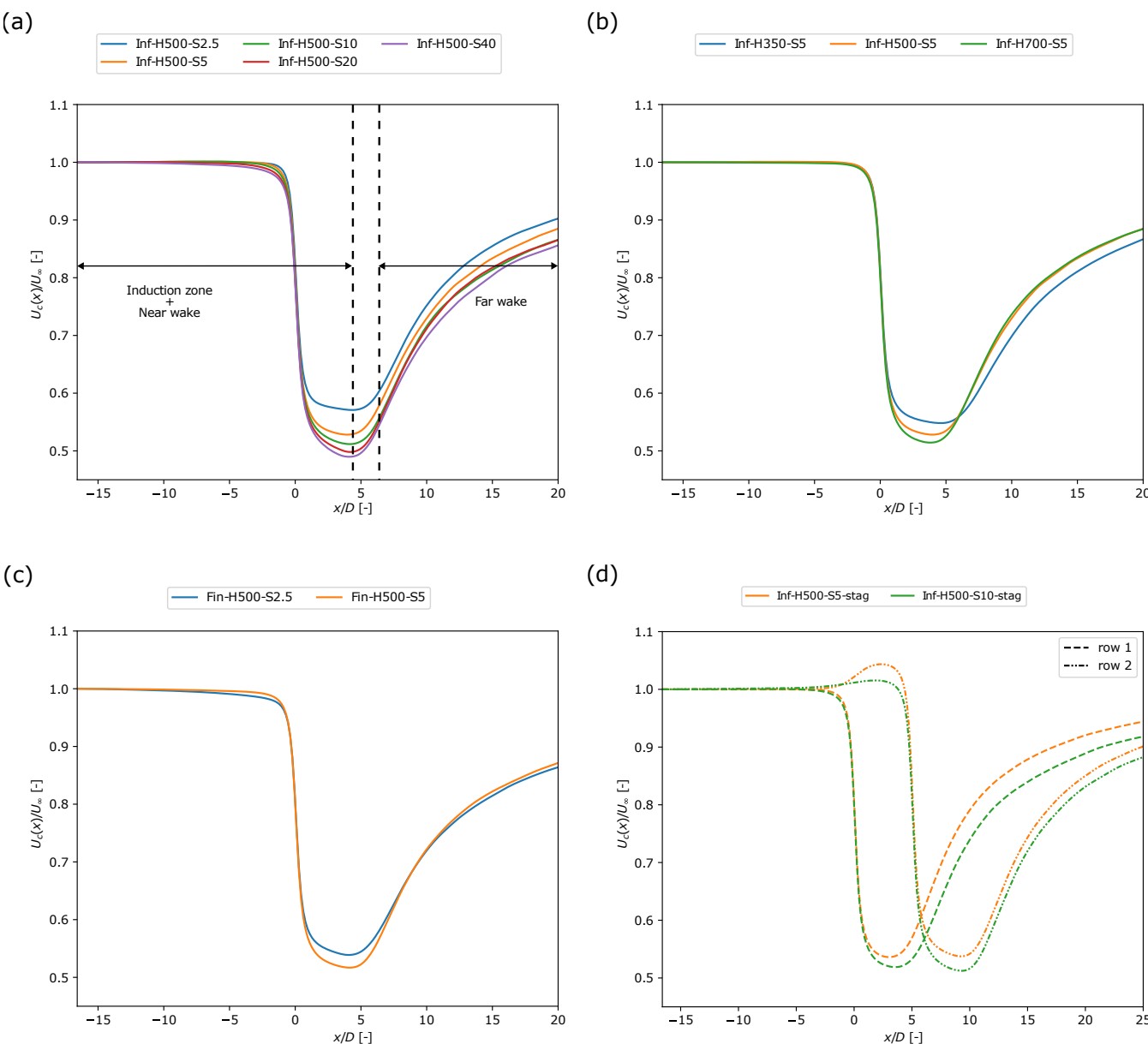

**Figure 4.** Streamwise centreline velocity $U_c$, scaled with the free stream velocity $U_\infty$, averaged over all turbines within the row for (a) single infinite row cases with $H = 500$ m, (b) single infinite row cases with $S = 5D$, (c) finite row cases and (d) staggered infinite row cases.

First looking at the near wake in Fig. 4, we observe that the maximum velocity deficit in the near wake decreases with $S$ and $H$, i.e. with increasing blockage, for all cases. Consequently, we also expect the axial induction factor ($a$) to depend significantly on blockage. In Fig. 5 we show the axial induction factor for all cases, as obtained from the LES, as function of the inverse geometrical blockage ratio ($A/A_d = 4SH/\pi D^2$), i.e. cases with strong blockage have a low inverse geometrical blockage ratio. We note that an alternative geometric blockage parameter based on the cross-section of the domain, i.e., $A_{domain}/A_{farm} = 4L_y H/N\pi D^2$, with $L_y$ the domain width, as, e.g. used for analysis of tidal channels (see, e.g., Nishino and Willden (2012)) is not meaningful here, since for our finite wind-farm simulations we are considering $L_y \to \infty$. We evaluate $a = 1 - U_d/U_{in}$, with $U_d$ the disk averaged (and time averaged) velocity, and $U_{in}$ the turbine inflow velocity at hub height. For all infinite row cases, we average in addition over the different turbines in the row (since they have in principle the same induction factor); for the finite row cases, we show the induction of the individual turbines. For all cases, except the staggered row cases, $U_{in}$ is simply the far upstream inflow velocity at hub height $U_\infty$, obtained from the concurrent precursor simulation. It is calculated as the streamwise velocity averaged along the streamwise line passing through the turbine hub location. For the staggered cases, in particular the second row, we observe however a significant acceleration of the flow upstream of the second-row turbines, which passes in between the turbines of the first row (see Fig. 4d). Therefore, for the second row, $U_{in}$ is defined as the maximum velocity upstream of the turbine, which is located at the end of the near wake of the turbines in the first row (see Fig. 4d). Looking in particular at the comparison between the axial induction factor in the first and second row of the staggered row case and single infinite row case, for same boundary layer height (and thus same inverse geometrical blockage ratio $A/A_d = 4SH/\pi D^2$), we observe that they are nearly equal and within averaging uncertainty, indicating that our choice of $U_{in}$ works well. Later (see Sect. 4) we will show that this definition also leads to good performance of the simple model (Eqs. 4–8) when compared to LES data.

The error bars in Fig. 5 are constructed using moving block bootstrapping (MBB). Time averaging was performed over a time interval $T_{av} = 10000$s, sampling every 2 seconds. The MBB method splits the original time series with $n$ data samples into $N_b = n - L + 1$ overlapping blocks containing $L$ samples. From this pool of $N_b$ blocks, a new time series is assembled by randomly choosing $K = n/L$ blocks with replacement and then the mean of the new time series is calculated. This process is repeated $B$ times, resulting in a distribution of means. Finally the 2.5 % and 97.5 % percentiles of the distribution of means mark the 95 % confidence interval. The MBB approach is defined by selecting the number of bootstrapping runs B and the size of the blocks $L$. The preliminary sensitivity study showed convergence, for $L = 20$, to a robust value that does not significantly change for longer block lengths. A similar sensitivity study showed that $B = 1500$ iterations is sufficient for this purpose.

Looking further at Fig. 5a, we observe that the axial induction factor converges to $a \approx 0.264$ for high inverse geometrical blockage ratio ($A/A_d$ high). This is inline with the expected Betz–Joukowsky value of $a = C_T'/(4 + C_T') = 0.265$. The plot also shows, that the finite row cases have an overall higher induction than their infinite row counterparts for the same inverse geometrical blockage ratio, implying that less blockage is present in the finite row cases. This will be explained further in Sect. 4.2.2. Overall, at low inverse geometrical blockage ratios ($A/A_d$ low), we see a significant deviation of the induction factor towards lower values than the expected Betz–Joukowsky value. This difference further translates towards the far wake as seen before in Fig. 4. It is further interesting to look at the induced near wake velocity. In Fig. 5b, we show $(1 - U_w/U_{in})/(2a)$,

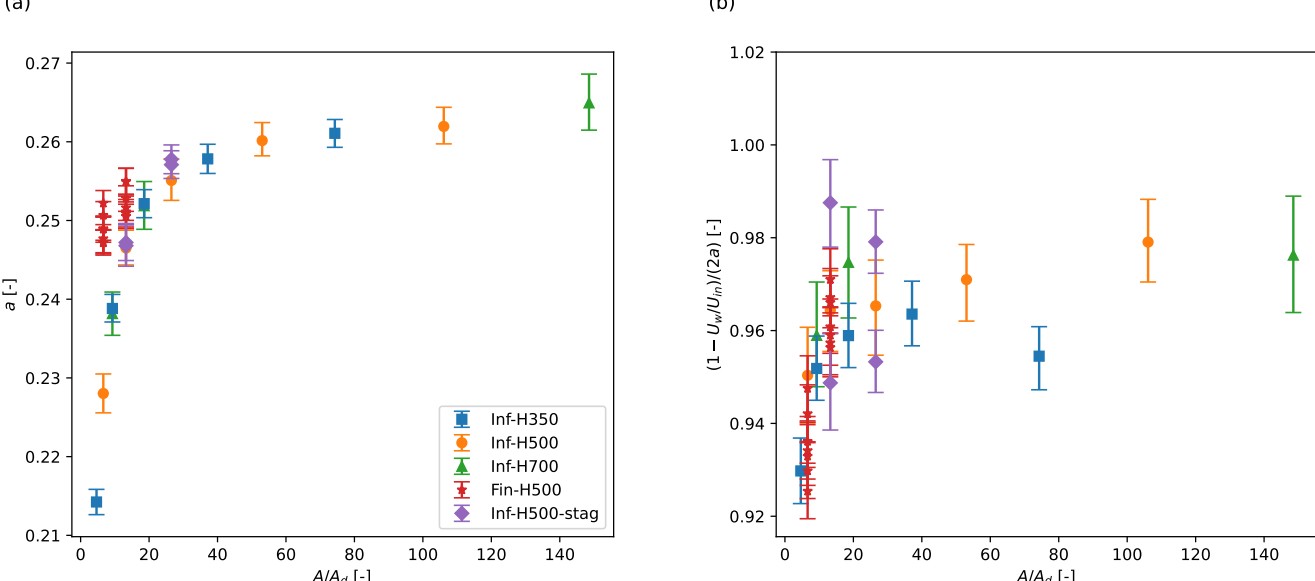

**Figure 5.** LES (a) axial induction factor $a$ and (b) induced near wake velocity scaled with the axial induction obtained, as function of the inverse geometrical blockage ratio $A/A_d = 4SH/(\pi D^2)$. The row averages are shown for the infinite row cases, while individual turbine values are shown for the finite row cases. Error bars represent 95% confidence intervals, obtained using moving block bootstrapping.

where $U_w$ evaluated as the minimum wake velocity (see Fig. 4) at hub height. Note that the expected Betz–Joukowsky value

of this ratio corresponds to one. As can be seen in Fig. 5b, the induced near wake velocity nears the Betz–Joukowsky value, for high inverse geometrical blockage, but some differences remain. In particular the onset of wake recovery can reduce the maximum wake deficit expected from the pure inviscid solution predicted by the Betz–Joukowsky. Next to that, subtle effects, related to the presence of shear may also play a role. When looking at decreasing inverse geometric blockage ratio's, we observe that not only a lower induction results at the turbine disk, but an even lower wake deficit in the near wake (i.e.

$(1 - U_w/U_{in}) < 2a)$.

### 4.1.2 Far wake analysis

We first discuss the evolution of the far wake as seen in a horizontal plane at hub height. Here we observe that a classical Gaussian shape function provides good fits along the downstream direction. This is in line with Liu et al. (2002), Shamsoddin and Porté-Agel (2017, 2018), who observed that the their turbulent wake profiles retained a Gaussian shape under non-zero

pressure gradient conditions. Thus we use a Gaussian profile (Pope, 2000),

$$\frac{U_s(x) - u(x,y)}{U_s(x)} = C(x)e^{-y^2/(2\delta_y(x)^2)}, \tag{9}$$

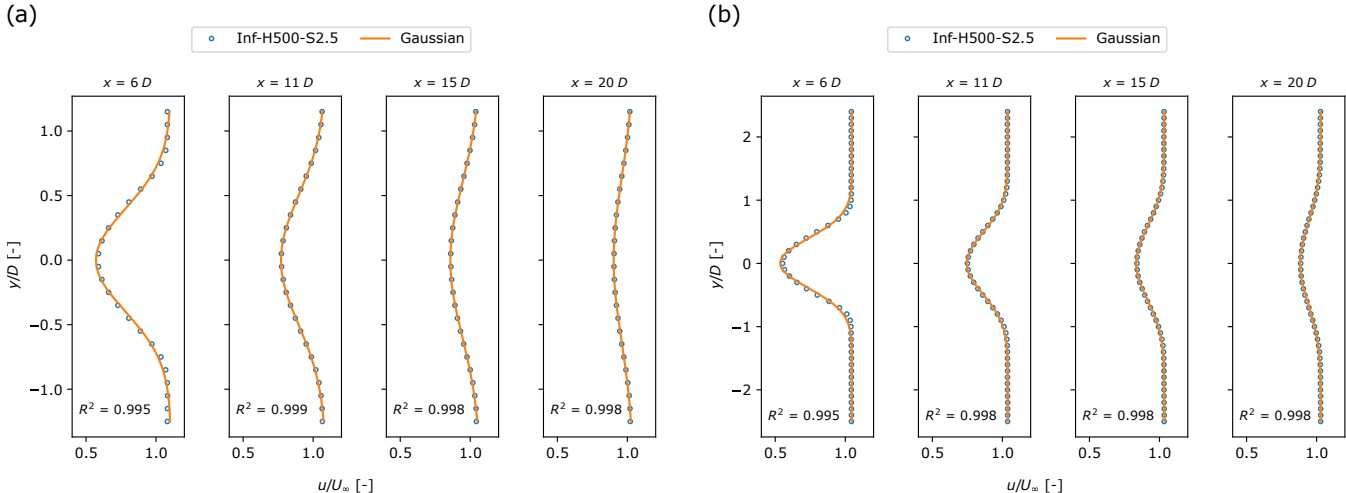

**Figure 6.** Horizontal wake velocity profiles, averaged over the turbines within the row, at different positions downstream in the far wake. The circles show LES results for the cases (a) Inf-H500-S2.5 and (b) Inf-H500-S5 and the orange lines represent the classical Gaussian shape function from Eq. 9. The $R^2$ value denotes the coefficient of determination.

where $U_s(x)$ is the streamwise velocity far from the wake centre, $u(x,y)$ is the wake velocity, $C(x)$ is the normalised wake velocity deficit and $\delta_y(x)$ is the horizontal wake width. We fit this function to the LES data, thus extracting $U_s(x)$, $C(x)$ and $\delta_y(x)$ as a result at each downstream position. Figures 6a and b, show indeed that the horizontal wake profile, averaged across turbines in the row, has a Gaussian shape in the specified downstream range. Similar profiles are also found for all the other simulation cases.

Figures 7a-c show the evolution of $C(x)$ and (d-f) show the evolution of $\delta_y(x)$, averaged over the different turbines in each row, scaled with their values at the start of the far wake for all simulations. The downstream distance is reformulated as $(x - x_0)/\delta_{y,0}$, with $x_0$ the starting location of the far wake. Figures 7b and e show that the far wake development of the two infinite staggered rows, as well as the single finite row, follow the same trend as the single infinite row. As $H$ increases, $C(x)$ shows a slightly steeper decline , while $\delta_y(x)$ spreads faster. In particular, the difference between $H = 350$ m and the other two BL heights is clearly visible. This can be understood by looking at the evolution of the vertical velocity profiles (in the wake centre) shown in Fig. 8 for infinite row cases with $S = 5D$. In particular for cases H350 is observed that vertical spreading of the wake is limited by the presence of the rigid-lid, while the maximum wake deficit seems minimally impacted. Note that the turbine tip height (270 m) is not much lower than the BL height in this case. Thus for this case, the turbine far wake behaves less as an axisymmetric and more as a planar wake, which is known to have a lower spreading rate (Pope, 2000). We remark that the rigid-lid only reduces the overall wake spreading at low $H$, while the wake recovery seems only slightly affected. We suggest that this imbalance is due to the unusual shape of the wake in the vertical direction for H350.

Returning to Fig. 7, we further observe that for constant $H$, the $C(x)$ and $\delta_y(x)$ curves align closely across varying $S$ values, and thus varying adverse pressure strengths. This contrasts with the findings of Liu et al. (2002); Shamsoddin and Porté-Agel

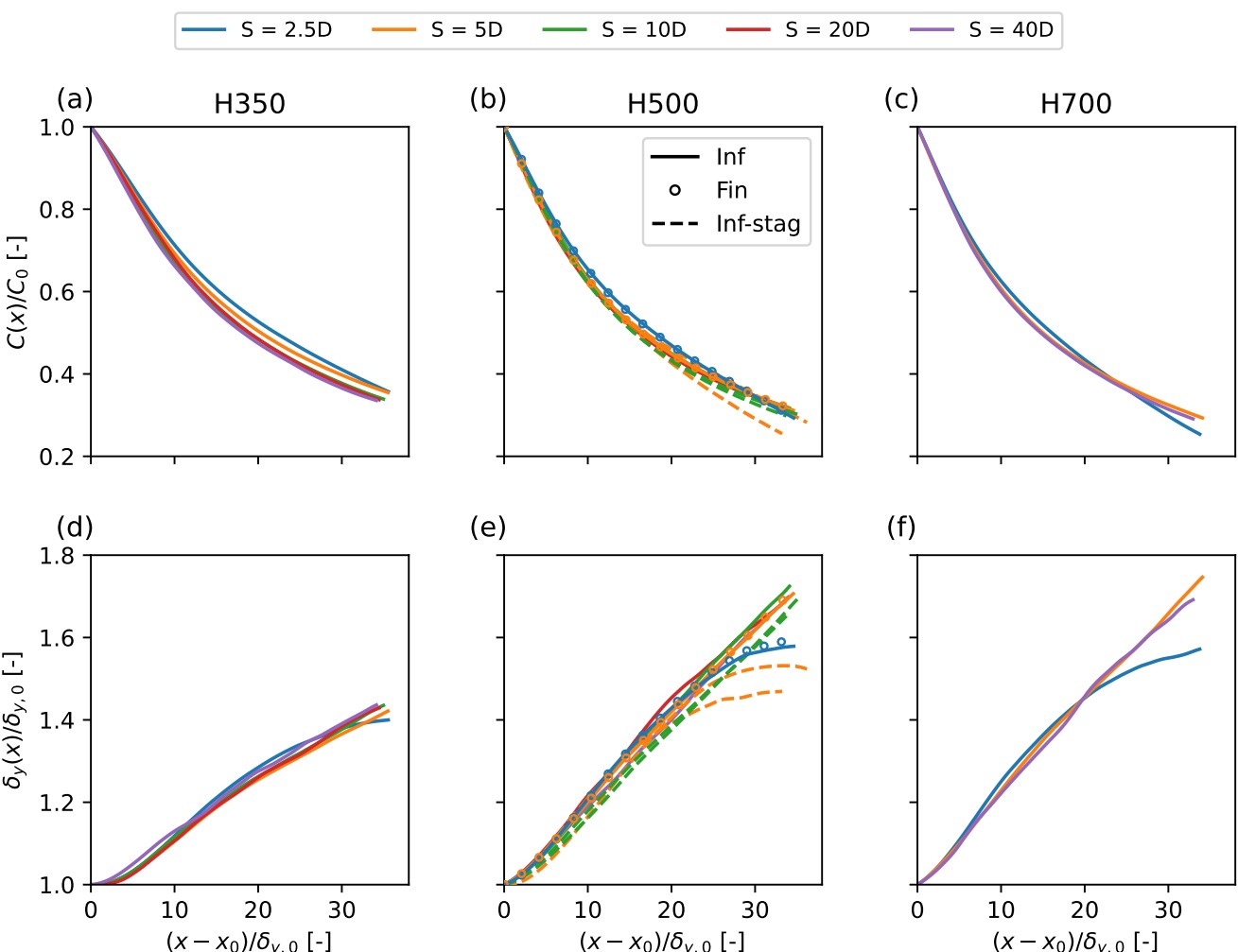

**Figure 7.** Evolution of the far wake properties for all simulations, averaged over the turbines within the row. (a–c) Normalised wake velocity deficit $C(x)$ and (d-f) wake width $\delta(x)$ scaled with their respective value at the start of the far wake $C_0 = C(x_0)$ and $\delta_{y,0} = \delta_{y,0}(x_0)$, for various $H$, plotted against the downstream distance $(x-x_0)/\delta_{y,0}$, with $x_0 = 6D$ the starting location of the far wake.

(2018), which suggest that a stronger adverse pressure gradient will slow down the wake deficit recovery and enhance wake spreading. However, the adverse pressure gradient in the far wake of our simulations (normalized by $\rho U_{in}^2$), turns out to be at least an order of magnitude smaller than the one in Liu et al. (2002) and Shamsoddin and Porté-Agel (2018), which they obtained as a result of diverging domain boundaries. Finally, looking back at Fig. 6a, which shows the turbine wake for the infinite row case Inf-H500-S2.5, which is periodic in the y-direction, we see no uniform flow at the sides of the wake, starting from $x = 15D$. This implies that for cases with $S = 2.5D$, (and also $S = 5D$ for the two staggered rows) neighbouring wakes start touching around this downstream location. This also drastically changes the wake spreading rate, as wakes essentially start

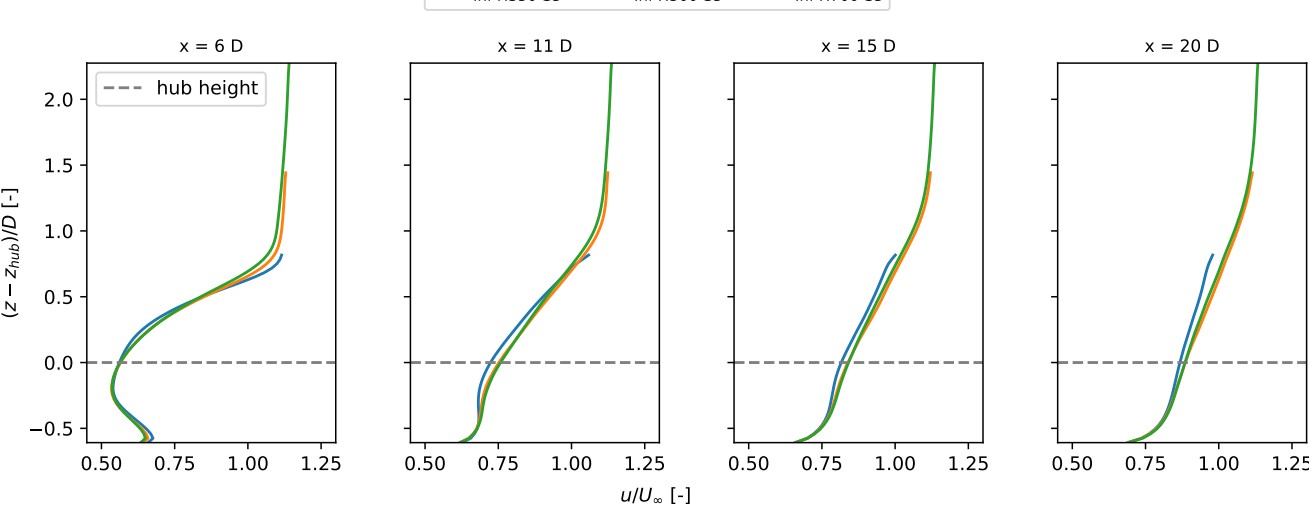

**Figure 8.** Vertical wake velocity profiles at different downstream positions, averaged over the turbines within the row for the infinite row turbine cases with $S = 5D$ and $H = 350, 500$ and $700$ m.

to merge. We thus conclude that blockage has negligible direct effect on the far wake recovery. Nevertheless, Fig. 4 shows a positive correlation between blockage and the near wake deficit. Therefore we suggest that blockage enhances wake efficiency, with the impact occurring primarily in the near wake.

### 4.2 Evaluation of near wake blockage model

We now focus on the effect of blockage on the power and thrust, and evaluate the simple near wake blockage model developed based on momentum theory (Eqs. 4–8) against the LES data. To this end, we evaluate from the large-eddy simulations

$$C_P = \frac{FU_d}{\frac{1}{2}A_d U_{in}^3} = C_T'\left(\frac{U_d}{U_{in}}\right)^3 = C_T'(1-a)^3, \tag{10}$$

$$C_T = \frac{F}{\frac{1}{2}A_d U_{in}^2} = C_t'\left(\frac{U_d}{U_{in}}\right)^2 = C_t'(1-a)^2, \tag{11}$$

where $a$ is evaluated as discussed above (see Sect. 4.1.1 and Fig. 5).

#### 4.2.1 Infinite row

For the infinite row cases, all input variables ($A_1 = A_2 = SH, A_d, C_t'$ and $U_{in}$) required for the blockage model are known. $U_{in}$ is obtained from the concurrent precursor as, the streamwise velocity averaged over the streamwise line through the turbine hub location, except in case of the second staggered row, where $U_{in}$ is derived from the wind farm simulation, as the maximum velocity upstream of the turbine. We also briefly compared with calculating $C_P$ and $C_T$ in the LES using a rotor-disk-averaged velocity at rotor height in the precursor domain. In the results (not shown here) this produced similar $C_P$ and $C_T$, with less

than a 1% difference for the highest blockage case. With $C_T'$ fixed, $C_P$ and $C_T$ depend on the ratio $U_d/U_{in}$ (i.e. the induction factor), which depends solely on the inverse geometrical blockage ratio. Thus for varying $U_{in}$, but $SH$ constant, the model returns a single $C_P$ and $C_T$ value.

Figures 9 (a) and (c), show $\langle C_P \rangle_{row}$ and $\langle C_T \rangle_{row}$ as function of $S/D$, scaled with $C_{P,0}$ and $C_{T,0}$, which are the $C_P$ and $C_T$ values of a single free standing turbine without blockage. To this end, we used the turbine in case Inf-H700-S40, since the geometric blockage ratio $A_d/A = 0.0067$, which can be regarded as negligible (Segalini and Inghels, 2014). As can be observed in the figure, the model agrees well with the LES data, although it tends to overpredict both thrust and power, in particular at high blockage ratios. We further see that at a turbine spacing of $S/D = 5$, there is a notable increase in $\langle C_P \rangle_{row}$ of 11%, 8% and 5% for $H = 350, 500$ and $700$ m, compared to the single free standing turbine.

Finally, Fig. 9b and d show $\langle C_P \rangle_{row}/C_{P,0}$ and $\langle C_T \rangle_{row}/C_{T,0}$ results from Fig. 9a and c, as a function of the inverse geometrical blockage ratio $(A/A_d)$. As mentioned above, the model output in this case only depends on the blockage ratio. We observe that the different LES results also collapse well onto a single curve when this scaling is used

### 4.2.2 Finite row

For the finite row cases, the known input variables are: $A_1 = SH$, $A_d$, $C_t'$ and $U_{in}$ which is obtained from the concurrent precursor as, the streamwise velocity averaged over the streamwise line through the turbine hub location. For this case, the area $A_2$ is not know a priori, nor is the favourable pressure gradient known. This would require an additional closure relation that related the wind-farm thrust to the larger pressure system around the farm.

To appraise this large scale pressure system, we evaluate for the finite row cases in Fig. 10a the pressure drop $-\Delta p_{nw}(y)$, only containing the pressure perturbations superimposed on the background pressure gradient (see Eq. 3). $-\Delta p_{nw}(y)$ is calculated as the difference between the domain inlet and the end of the near wake (defined as the location of maximum wake deficit), averaged over $H$, and shown as a function of the spanwise direction $y$, where $y = 0$ is located at the row centre. It is observed that the pressure drop reaches a maximum at the row centre, decreasing for larger absolute values of $y$ until a (near zero) pressure difference is reached far from the turbine row. As expected, the wakes of turbines at the edge of the row have more space for lateral expansion, and thus experience a lower $|\Delta p_{NW}|$ than those at the centre, where expansion is constrained by neighbouring turbines and the boundary layer height. This explains the overall lower axial induction found in the finite row cases compared to the equivalent infinite row cases, as seen in Fig. 5. All turbines in a finite row have less neighbouring turbines, resulting in a less constrained flow around the turbines. This can be considered as the collective blockage effect. Turbines within an infinite row are thus an extreme example of the centre turbines in a finite row. Additional simulations are needed to quantify the transition of a finite row into an infinite, but this is beyond the scope of the current study. Looking in more detail at the pressure drop around the turbines, we see small local variations. These are in part related to the normal Reynolds stresses. For a conventional axisymmetric wake it is well understood that $(1/\rho)\partial p/\partial r + \partial \overline{u_r' u_r'}/\partial r = 0$, with $\overline{u_r' u_r'}$ the radial normal stress (Pope, 2000). Thus, in Fig. 10b we have plotted $\Delta[p/\rho + \overline{v'v'}]$ as function of y (and averaged over the BL height). Note that the spanwise fluctuation $v'$ is strictly speaking only in the radial direction in a horizontal plane, but as appreciated in the figure the local variations in the pressure drop (in particular in the wake centres) disappear.

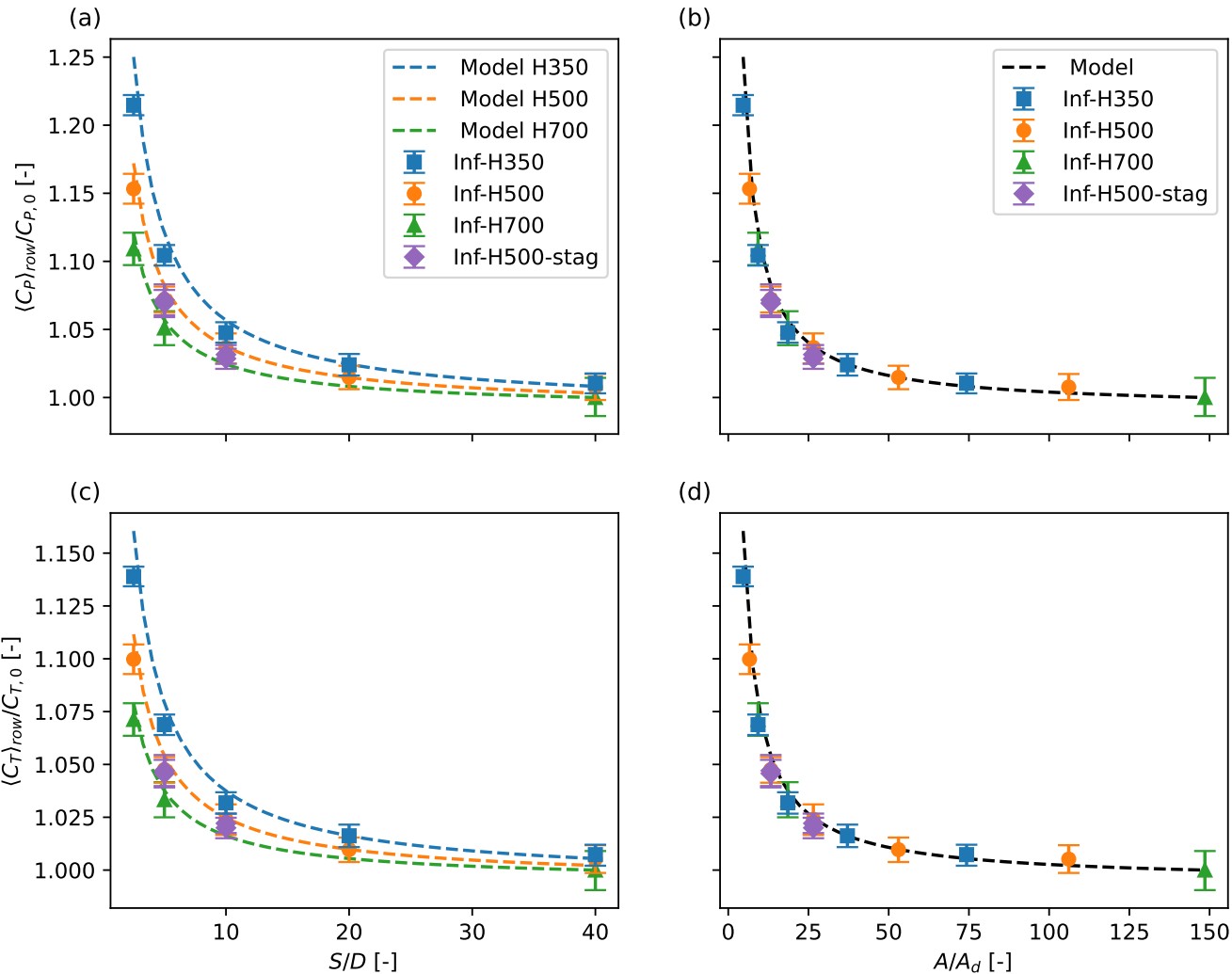

**Figure 9.** (a) Row averaged scaled power coefficient $\langle C_P \rangle_{row}/C_{P,0}$ as function of spacing $S/D$ and (b) as function of inverse geometric blockage ratio $A/A_d$, for varying boundary layer heights $H$. (c) Row averaged scaled thrust coefficient $\langle C_T \rangle_{row}/C_{T,0}$ as function of $S/D$ and (d) as function of $A/A_d$, for varying $H$. Data shown for all infinite row cases (single and staggered). $C_{P,0}$ and $C_{T,0}$ are the $C_P$ and $C_T$ of a single free-standing turbine with no blockage, i.e. case Inf-H700-S40. The error bars are bootstrap 95 % confidence intervals.

To close the system of equations (4–8), we will use $\Delta p_{NW}$ as observed in the LES, and compare the thrust and power output of the near wake blockage model with that of the LES. To obtain $\Delta p_{NW}$, we simply average $\Delta p_{nw}(y)$ over the interval $[y_h - S/2, y_h + S/2]$ for each turbine in the row. We note that a pressure closure at the farm level could, e.g., be provided by typical gravity wave models such as, e.g., WAYVE Allaerts and Meyers, 2019; Devesse et al., 2022; Stipa et al., 2024; Devesse et al., 2024a, b, in which a coupling between the larger atmospheric flow around the farm and local wake models is foreseen.

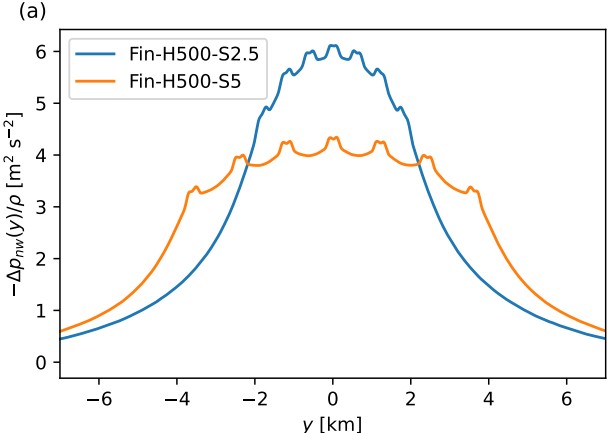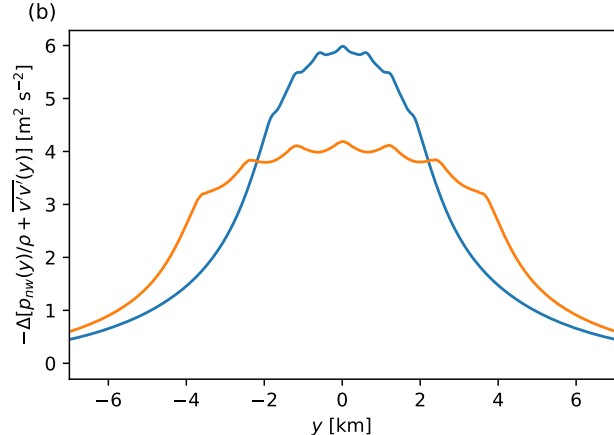

**Figure 10.** (a) Pressure drop at the end of the near wake ($-\Delta p_{nw}(y)$) and (b) pressure drop plus the lateral Reynold stress ($\overline{v'v'}(y)$), averaged over $H$, as function of the spanwise direction, where y = 0 km is the location of the centre turbine in the row.

However, here we aim specifically at assessing the relevance of a simple near wake model that accounts for a favourable pressure drop. A full interaction with effects of gravity-wave feedback is a topic of future research.

Next to closing the system with the pressure gradient from LES, we also evaluate the two-scale model proposed by Nishino and Willden (2012, 2013) for half-open channel flows (formulated in the context of tidal turbines). At the turbine scale (i.e. the near wake model), they use relations similar to (4–8), while at the farm scale, they use classical Froude momentum theory without pressure gradient to obtain a relation for the overall farm wake expansion. By further assuming that all turbines in a row produce the same power, they arrive at a closed system of equations. Although, the drawback of the approach is that there is no spanwise variation in turbine power output, it does allow for an overall assessment of farm power. We refer to Nishino and Willden (2012, 2013) for more details about this model and its implementation.

Figure 11a and b present the $C_P$ and $C_T$ for each turbine of cases Fin-H500-S2.5 and Fin-H500-S5, together with the predicted values from the extended Froude momentum theory model and the model developed by Nishino and Willden (2012). We notice that $\langle C_P \rangle_{row}$ is higher for the case with smaller $S$, i.e. stronger blockage, which was also observed for the infinite row cases. Additionally, the $C_P$ value increases towards the row centre, with $C_{P,centre} > \langle C_P \rangle_{row} > C_{P,side}$, which was also found in the RANS results of Nishino and Draper (2015). The same findings are observed for the $C_T$ data in Fig. 11b. We appreciate from the figures that for the high blockage case (S2.5) both models overpredict the power. For the lower blockage case (S5), Nishino and Willden's (2012; 2013) model is reasonably close to the row average predicted by the LES, whereas our model manages to predict also the shape of power and thrust along the row rather well.

Finally, in Fig. 12 the power and thrust coefficients $C_P$ and $C_T$, are provided for all simulations, assembling the results form Figs. 9 and 11 into one figure. Overall, the model agrees well with the LES data, although it tends to overpredict both thrust and power, in particular at high blockage ratios, for both the infinite and finite row cases the error increases. We observe smaller

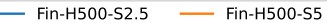

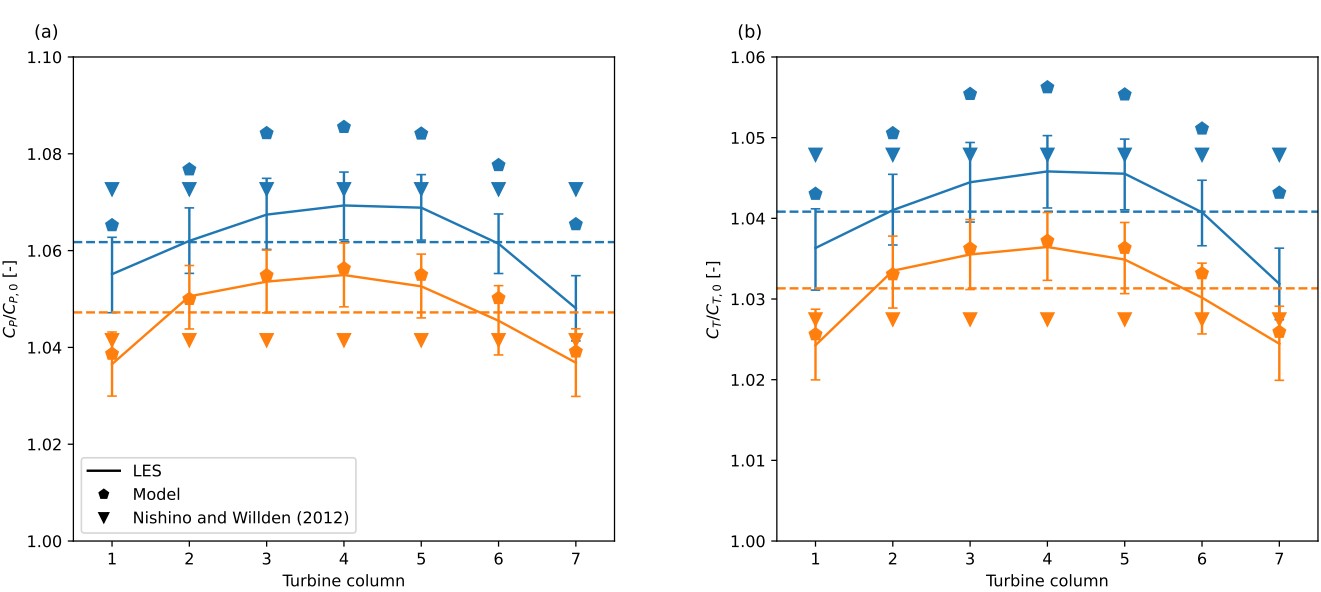

**Figure 11.** (a) $C_P$ and (b) $C_T$ for each turbine in the finite row cases, as found by the large-eddy simulations, the extended Froude momentum theory model and the model from Nishino and Willden (2012). The error bars are bootstrap 95 % confidence intervals. Horizontal dashed lines mark the row averaged values of the corresponding LES cases.

increases in $C_P$ and $C_T$ for the finite row cases, compared to their infinite row counterpart, since less surrounding turbines are present to constraint the inflow from expanding, thus resulting in weaker pressure drops and blockage.

### 4.3 Discussion

To locate the origin of the model error, we perform a streamwise momentum budget analysis. The budget equation is derived by time-averaging the streamwise momentum equation and then integrating it over the control volume $\Omega$, which is the same as the model's control volume. In the streamwise direction, the control volume extends from the computational domain entrance, to the end of the near wake (defined as the location with maximum wake deficit), $x_1 = 0$ to $x_2 = x_{NW}$. The vertical dimension of the control volume coincides with the vertical boundaries of the computational domain, that is from $z_1 = 0$ to $z_2 = L_z$.

Therefore there is no momentum flux across those faces. The lateral extension of the control volume follows the streamlines originating from the domain entrance at locations $y = y_{hub} \pm S/2$. Therefore $(\bar{u}n_x + \bar{v}n_y) = 0$ and thus no advection of momentum across those faces (See Appendix A, Fig. A1 for a visualization of the streamlines). We denote $x_1, x_2, s_1, s_2, z_1$ and $z_2$ as the boundaries of the control volume, where $s$ represents the streamline coordinate. The y-z and x-y boundary faces are denoted by $A_x$ and $\Psi_z$ respectively, while the s-z streamline boundary faces are denoted by $\Gamma_s$. By applying the divergence

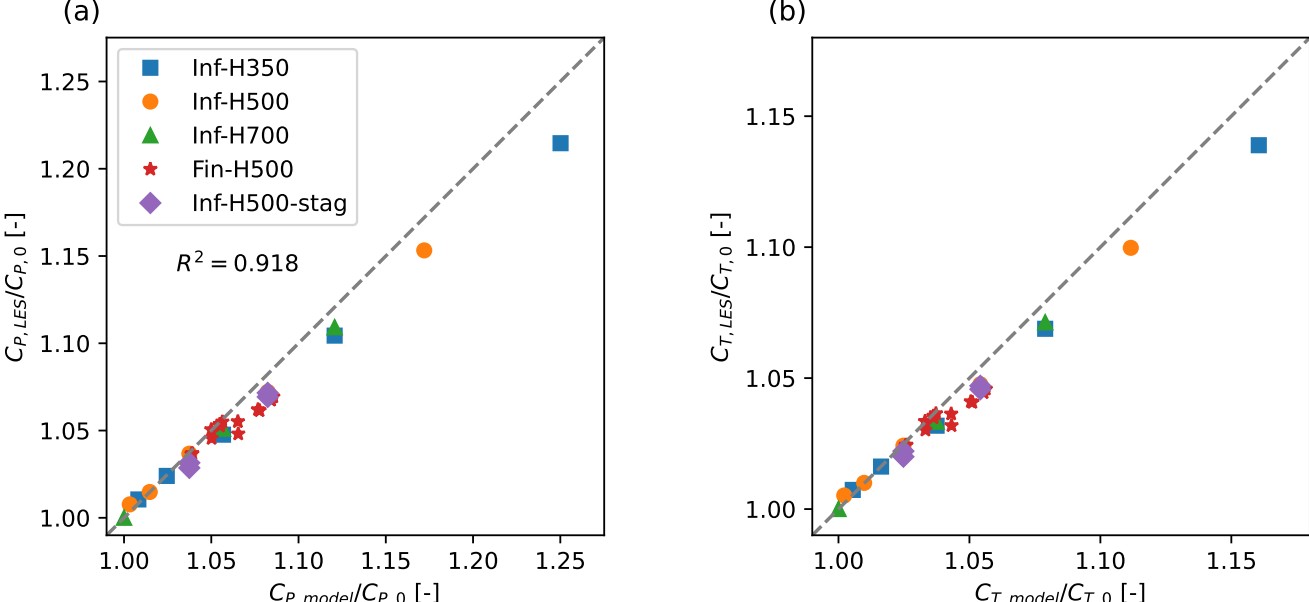

**Figure 12.** (a) The scaled power coefficient $C_{P,LES}/C_{P,0}$ and (b) The scaled thrust coefficient $C_{T,LES}/C_{T,0}$, as found by the extended Froude momentum theory model (x-axis) and the large-eddy simulations (y-axis). The squares, circles and triangles represent the row averaged values of the single infinite row cases, with respective $H$. The stars represent the values of the individual turbines for the finite row cases. The diamonds represent the row averaged values of the infinite staggered row cases. The $R^2$ value denotes the coefficient of determination.

theorem on the streamwise momentum equation, the divergence operator is removed, allowing the conversion of the volume integral into a surface integral. The resulting equation is,

$$
\underbrace{-\left[\int_{A_x} \bar{u}\bar{u}\, dA_x\right]_{x_1}^{x_2}}_{\Delta \mathcal{M}_{A_x}} \underbrace{-\left[\int_{A_x} \left(\overline{u'u'}^r + \tau_{xx}^{\mathrm{sgs}}\right) dA_x\right]_{x_1}^{x_2}}_{\Delta \mathcal{F}_{A_x}} + \underbrace{\sum_{j=1}^{2}\left[\int_{\Gamma_s} \left(\left(\overline{u'u'}^r + \tau_{xx}^{\mathrm{sgs}}\right)n_x + \left(\overline{u'v'}^r + \tau_{xy}^{\mathrm{sgs}}\right)n_y\right) d\Gamma_s\right]_{/}^{s_j}}_{\Delta \mathcal{F}_{\Gamma_s}}
$$

$$
\underbrace{+\left[\int_{\Psi_z} \tau_{xz} n_z\, d\Psi_z\right]_{/}^{z_1}}_{\Delta \mathcal{F}_{fric}^*} \underbrace{-\int_{\Omega} \frac{1}{\rho_o}\frac{\partial \bar{p}_\infty}{\partial x}\, d\Omega}_{} \underbrace{-\int_{\Omega} \frac{1}{\rho_o}\frac{\partial \bar{p}^*}{\partial x}\, d\Omega}_{\mathcal{P}_x^*} + \underbrace{\int_{\Omega} \bar{f}_x\, d\Omega}_{\mathcal{F}_{t,x}} = 0,
\tag{12}
$$

where $u' = u - \bar{u}$,

$$\underbrace{\int_\Omega \frac{1}{\rho_o} \frac{\partial \bar{p}^*}{\partial x} d\Omega}_{\mathcal{P}_x^*} = \underbrace{\left[ \int_{A_x} \frac{1}{\rho_o} \bar{p}^* dA_x \right]_{x_1}^{x_2}}_{\mathcal{P}_{A_x}^*} + \underbrace{\sum_{j=1}^2 \left[ \int_{\Gamma_s} \frac{1}{\rho_o} \bar{p}^* n_x d\Gamma_s \right]^{s_j}}_{\mathcal{P}_{\Gamma_s}^*} . \tag{13}$$

The term $\Delta\mathcal{M}_{A_x}$ describes the advection of streamwise momentum by the mean streamwise velocity, while $\Delta F_{A_x}$ accounts for the total (both resolved and modelled) turbulent transport of streamwise momentum along the streamwise direction. Further, $\Delta\mathcal{F}_{fric}^*$ captures the additional ground friction, not balanced by the background pressure gradient, due to the side flow acceleration induced by the turbines. Lastly, $\mathcal{P}_x^*$ accounts for pressure gradients caused by wind farm effects and $\mathcal{F}_{t,x}$ stands for the turbine thrust force. By the sign convention used, the first six terms in eq. (12) are positive when the flow or turbulence brings more momentum into the control volume than out, and negative when the opposite occurs. For instance, $\Delta\mathcal{M}_{A_x}$ is positive if the mean streamwise momentum integrated over $A_{x_1}$ is greater than over $A_{x_2}$. An increase in momentum along x would make this term negative.

The terms in the above equation are calculated and compared in Fig. 13 for the LES simulations and the model. We remark that in the model, the terms $\Delta\mathcal{F}_{A_x}, \Delta\mathcal{F}_{\Gamma_s}$ and $\Delta\mathcal{F}_{fric}^*$ are not represented and therefore equal to zero. Figure 13a shows the comparison of the terms, between the model and the LES for cases Inf-H350-S2.5 and Inf-H500-S2.5, scaled with the momentum flux going into the domain $\mathcal{M}_{A_1}$. It is clear that the model is capable of representing the dominant terms ($\Delta\mathcal{M}_{A_x}, \mathcal{P}_x^*$ and $\mathcal{F}_{t,x}$) in the momentum equation. More specifically, as also shown in Fig. 13b, the model can accurately predict the thrust, yet overpredicts the magnitude of $\Delta\mathcal{M}_{A_x}$ and $\mathcal{P}_x^*$. This error increases for higher blockage cases. We identify 3 main causes for the overprediction of the model. First, the overprediction may partly be due to the model not accounting for the smaller remaining terms $\Delta\mathcal{F}_{A_x}, \Delta\mathcal{F}_{\Gamma_s}$ and $\Delta\mathcal{F}_{fric}^*$. For the infinite row cases, $\Delta\mathcal{F}_{\Gamma_s}$ is expected to be zero, since the control volume is not expanding (see Fig. A1a in appendix A) , therefore the in- and outflowing turbulent momentum should be equal at the side boundaries. The term $\Delta\mathcal{F}_{A_x}$ could be added to the pressure term $\mathcal{P}_x^*$ if $\overline{u'u'}$ has no direct effect on the thrust. Moreover, $\Delta\mathcal{F}_{fric}^*$ is stated to be zero in the model, as we assumed that the ground friction is fully balanced by the background pressure gradient. The LES shows that the ground friction increases with the blockage strength and therefore $\Delta\mathcal{F}_{fric}^*$ is not zero, although the term is small. Secondly, the overprediction may be because of the poor representation of the near wake velocity profile. The model assumes a top-hat function whereas the LES already exhibits a smoothed top-hat profile. Therefore, the model will overpredict the streamwise momentum flux $\mathcal{M}_{A_2}$ at the end of the near wake. Doubling the grid resolution of the LES in all three directions, as done for case Inf-H350-S2.5 in Fig. 13a and b, produces similar results. This suggests that within the current range of grid resolutions, the filter width of the actuator disk model has an insignificant effect on the near wake shape. Thus, we conclude that the smoothed near wake velocity profile is prominantly a result of turbulent mixing in the near wake. In future work, the model could be improved by defining the near wake velocity profile with a shape function, and parametrizing the near wake energy as a function of turbine thrust and ambient turbulence. Lastly, the overprediction could be due to the Shapiro correction. The correction was developed for a single turbine with no blockage. LES without the Shapiro correction shows that the correction introduces additional uncertainty for the highest blockage case ($\approx 8\%, 6\%$ and $4\%$ in-

crease for $\Delta\mathcal{M}_{A_x}, \Delta\mathcal{P}_x^*$ and $\Delta\mathcal{F}_{t,x}$ compared to the case without Shapiro correction). In a future study, the turbines could be simulated using the actuator line model instead of the actuator disk model, this would circumvent the need for the correction.

Finally, we look at the finite row case Fin-H500-S2.5 in Fig. 13c and d. The conclusions drawn for the infinite row cases also hold for the finite row cases. We remark that now the model error on $\mathcal{P}_x^*$ is drastically smaller, because $\Delta p_{NW}$ obtained from the LES is directly used as an input to the model. However the error on this term is not zero because the model assumption of the side pressure $p_{side} = p_{in} + \alpha\Delta p_{NW}$ with $\alpha = 0.5$, is not entirely correct. The LES shows that $p_{side}$ shifts closer to $p_{in}$, i.e. $\alpha < 0.5$, however $\alpha$ depends on the blockage strength. Nevertheless, the contribution of $p_{side}$ in the streamwise momentum balance is significantly smaller than the other terms, due to the small streamwise projection of the side surfaces.

## 5   Conclusions

This study set out to analyse the effect of blockage on the wake development behind turbines and the turbine power. Recent research by Lanzilao and Meyers (2024) discovered a strong positive correlation between the favourable pressure gradient in the farm and the wake efficiency $\eta_w$. This implies that the favourable pressure gradient, induced by blockage, enhances the wake recovery mechanism. We performed 17 LES simulations consisting of infinite and finite single turbine rows, as well as two staggered turbine rows, with constant $C_T'$, in an idealised ABL setting. Blockage conditions were artificially introduced using a rigid lid, inducing a favourable pressure difference ($\Delta p_{NW} < 0$) over the turbine row and an adverse pressure difference ($\Delta p_{FW} > 0$) in the far wake. The blockage strength was adjusted by varying the turbine spacing ($S/D = 2.5, 5, 10, 20, 40$) and boundary layer height ($H = 350, 500, 700$ m).

A strong positive correlation was identified between $\Delta p_{NW}$ and both the power coefficient ($C_P$) and thrust coefficient ($C_T$). Specifically, as $S$ and $H$ decrease, $-\Delta p_{NW}$, $C_P$, and $C_T$ increase. Simultaneously, the rotor disk experiences lower induction, and the near wake shows a reduced wake deficit. Specifically, the infinite row cases, at realistic spacings $S/D = 5$ already show a significant increase in $C_P$ of 11%, 8% and 5% for $H = 350$, 500 and 700 m respectively, compared to a single free standing turbine without blockage. Smaller increases are found for the finite row cases, since less surrounding turbines are present to constraint the inflow from expanding, thus resulting in weaker pressure drops and blockage. In this case, power and thrust are distributed, are maximum at the centre of the row, and also $-\Delta p_{NW}$ increases towards the centre of the row.

The reduction in near wake velocity deficit due to blockage also leads to smaller velocity deficits and narrower wake widths in the far wake. However, blockage has a negligible direct effect on far wake development when scaling the far wake deficit and width with their initial far wake values. We note that the adverse pressure gradient in the far wake that we observed in our study, is at least an order of magnitude smaller than those found by, (Liu et al., 2002; Shamsoddin and Porté-Agel, 2018) in diverging channels (for which case adverse effects on wake recovery were noticed). We thus conclude that the blockage enhances the wake efficiency, an effect that originates primarily from the lower near wake deficit. However, we do see a profound effect of $H$ on the wake spreading, with higher boundary layers leading to faster spreading. This relates to the fact that the wake can more freely expand vertically in high-boundary layer cases, into a larger region of high-speed flow than for shallow boundary layers.

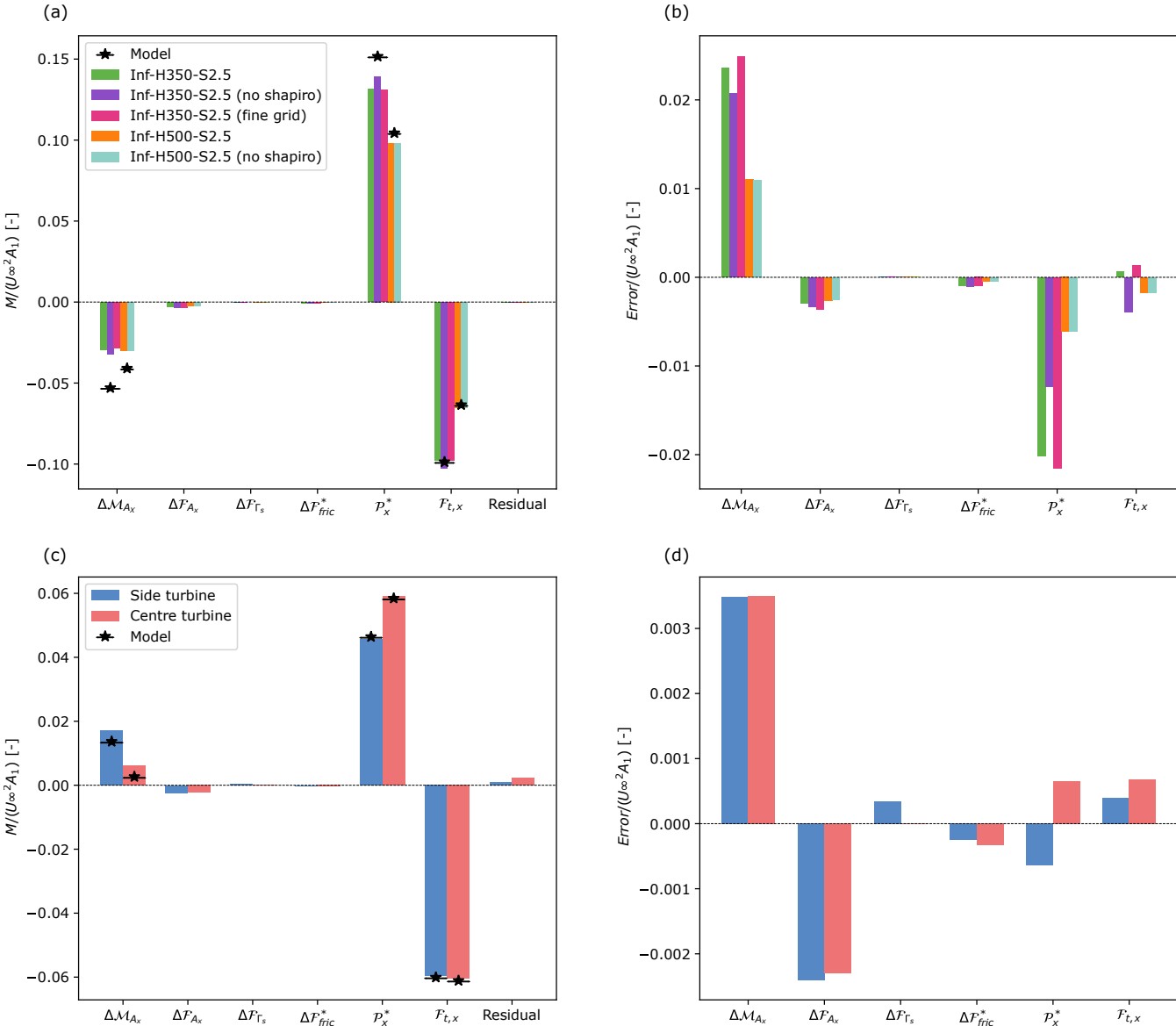

**Figure 13.** Comparison of the momentum sources and sinks as calculated by eq. (12), between the developed model and the LES cases. (a) Row averaged results of Inf-H350-S2.5 standard case, without shapiro correction and a finer grid, as well as Inf-H500-S2.5 standard case and without shapiro correction. (c) Side and centre turbines of Fin-H500-S2.5. (b) and (d) show the corresponding model errors.

We further developed an analytical model consisting of five equations to predict the blockage effect on near wake properties (similar to earlier models to correct for wind tunnel blockage in experiments – see Mikkelsen and Sørensen, 2002; Werle, 2010; Segalini and Inghels, 2014). Based on theoretical Froude momentum theory applied to a confined domain, the model uses four known input variables: rotor diameter ($D$), turbine spacing ($S$), boundary layer height ($H$), disk based thrust coefficient ($C'_T$)

and inflow velocity $(U_{in})$, but requires one additional closure relation. In the infinite row case, this is given by $A_1 = A_2$ (no expansion possible at farm level), whereas in the finite row case, the near wake pressure drop needs to be provided. To this end, we provided the pressure drop as measured in the large-eddy simulations. Overall, we found a good agreement between LES power and thrust predictions and the simple model based on momentum theory, indicating that the latter is a suitable candidate to improve turbine power prediction under blocking conditions. However the model does underperform for the very high blockage cases. A momentum budget analysis identified the sources of the model overprediction, which can be improved primarily by parametrizing the near-wake velocity profile using a shape function, on the one hand, and possibly by parametrizing the ground friction, on the other hand. Furthermore, more detailed validation using LES using in combination with an actuator line model, so that uncertainties related to the Shapiro correction can be excludes, are also of interest. In a practical implementation, such a model needs to be coupled to a farm-scale model that provides input on the large pressure distribution in the farm. Atmospheric perturbation models (Allaerts and Meyers, 2019; Devesse et al., 2022; Stipa et al., 2024; Devesse et al., 2024b; see also the open-source model WAYVE) that explicitly model the pressure feedback coming from gravity waves, are an interesting application for this. This is an ongoing topic of further research.

## Appendix A: Control volume streamlines

This appendix aims to visualize the control volume around the turbines, as used in both the developed model and the momentum budget analysis. The control volume is bounded by streamlines derived from the time-averaged velocity fields of the LES, extending from the domain inlet to the end of the near wake region. Figures A1a and b illustrate these streamlines at hub height for the infinite and finite row cases. For clarity, only four turbines are shown in each case. In the finite row case, these include the side turbine up to the row centred turbine. We remark that the assumption $A_1 = A_2$ holds for the infinite row case, while $A_2 > A_1$ is valid for the finite row case.

*Code availability.* The Navier–Stokes solver used in this work is SP-Wind, a proprietary software with restricted access. Access may be granted upon reasonable request.

*Data availability.* The full dataset generated during the study is available from the corresponding author upon reasonable request. The data and the Python scripts necessary to reproduce the figures in this study are openly available as a KU Leuven RDR dataset: dataset: https://doi.org/doi:10.48804/INE8OG (Olivier, 2025).

*Author contributions.* ON and JM jointly defined the methodology and set up the simulation studies in the current work. ON, JM and AP jointly developed the analytical models. ON carried out the simulations and post-processing. ON and JM wrote the manuscript.

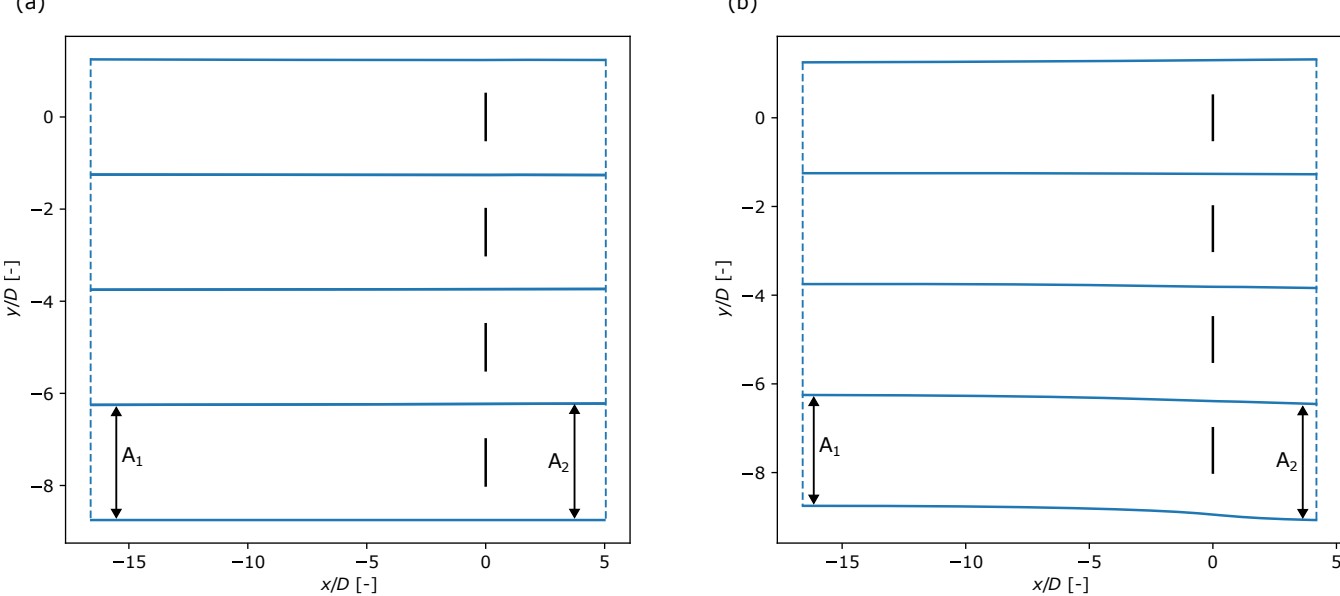

**Figure A1.** Plan view at hub height of four turbines in (a) case Inf-H500-S2.5 and (b) case Fin-H500-S2.5. The solid blue lines are the streamlines calculated from the LES. The blue striped lines mark the beginning and end of the control volumes. The vertical black lines represent the turbines. The turbine at the centre of the finite row has coordinates $(x/D, y/D) = (0,0)$ and at the edge $(x/D, y/D) = (0, -7.5)$.

*Competing interests.* At least one of the (co-)authors is a member of the editorial board of Wind Energy Science. The authors have no other competing interests to declare.

*Acknowledgements.* The computational resources and services used in this work were provided by the VSC (Flemish Supercomputer Center), funded by the Research Foundation - Flanders (FWO) and the Flemish Government. Additionally, the authors express their gratitude to the anonymous reviewers for their insightful comments and relevant observations on this work.

*Financial support.* Project Cloud4Wake, funded by Vlaamse Agentschap Innoveren & Ondernemen (VLAIO) under Blue Cluster cSBO programme (Contract number HBC.2022.0549)

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
