# Peer review of "Effect of blockage on wind turbine power and wake development"

_Wind Energy Science, 2025_

## Author Comment (AC1)

**Response to the reviewers**

We thank the reviewers for their critical assessment of our work. In the following we address their concerns point by point which has helped to improve our manuscript. The reply below gives a motivation of the changes, as well as the additions to the manuscript (*in italic*). The figure numbers in the reply refer to the figures in the improved manuscript.

**Reviewer 1**

In this work, the authors conducted 17 idealized large-eddy simulations (LES) with different blockage conditions by varying turbine spacing and boundary layer height to investigate the impact of blockage on wake development and power output of wind turbines. Additionally, the authors developed an analytical model based on the classic Froude momentum theory to predict the blockage effect on near-wake characteristics and compared the results obtained from the analytical model with the LES results. The work is well-conducted and of interest to the community. Some minor comments are as follows:

**Reviewer Point P 1.1** — The causes for the discrepancies shown in Fig. 11 need to be better explained, especially considering that the LES results are already employed for model closure.

**Reply**: We thank the reviewer for this comment. We have added an additional discussion section in the updated manuscript (see Sect. 4.3), where we investigate the model error through a detailed momentum budget analysis.

**Reviewer Point P 1.2** — The order of Eqs. 4 and 5 does not match their order in the corresponding text.

Reply: This has been updated in the manuscript.

**Reviewer Point P1.3** — Lines 180–181: "is,er" appears to be a typographical error.

**Reply**: This has been updated in the manuscript.

**Reviewer Point P 1.4** — The abscissa in Fig. 6 has dimensions, while it is nondimensionalized in Fig. 8. Ensure consistency throughout the paper.

**Reply**: Figures 6 and 8 have now been changed in the updated manuscript, such that all abscissa are nondimensionalized. Specifically, in Fig. 6 the x-axis is changed from u to  $u/U_{\infty}$ , and in Fig. 8 the y-axis is changed from z to  $(z - z_{hub})/D$ .

**Reviewer Point P 1.5** — Lines 278–279: "Returning to Fig. 7, we further observe that for constant H, the C(x) and  $\delta_y(x)$  curves align closely across varying S values, and thus varying adverse pressure strengths." This is not the case in Fig. 7(f) at far-wake locations.

**Reply**: We thank the reviewer for this observation. Indeed, in Fig. 7f, the second half of the farwake of cases Inf-H700-S5 and Inf-H700-S40 do not overlap. This was a mistake in the post processing of this figure. When making Fig. 7c and f for the Inf-H700 cases, we accidentally selected the cases

without shifted periodic boundary conditions. These cases still have large streamwise turbulent structures (approximately the same size as the boundary layer height), present in the flow, which will effect the shape and recovery of the turbine wakes, located inside those structures. This additional uncertainty due to the large turbulent structures, caused the deviation in the second half of the the far wake. Figure 7c and f are now updated in the manuscript with the Inf-H700 cases including shifted periodic boundary conditions (see Fig. RR1).

Figure RR1: Evolution of the far wake properties for all simulations, averaged over the turbines within the row. (a–c) Normalised wake velocity deficit C(x) and (d-f) wake width  $\delta(x)$  scaled with their respective value at the start of the far wake  $C_0 = C(x_0)$  and  $\delta_{y,0} = \delta_{y,0}(x_0)$ , for various H, plotted against the downstream distance  $(x - x_0)/\delta_{y,0}$ , with  $x_0 = 6D$  the starting location of the far wake.

**Reviewer Point P 1.6** — Lines 283–284: "Finally for cases with S = 2.5D, (and also S = 5D for the two staggered rows) neighbouring wakes start touching at  $x \approx 16D$  (see Fig. 6a)."Please verify if the authors are referring to Fig. 6a.

**Reply**: This has been clarified in the updated manuscript:

Finally, looking back at Fig. 6a, which shows the turbine wake for the infinite row case Inf-H500-S2.5, which is periodic in the y-direction, we see no uniform flow at the sides of the wake, starting from x = 15D. This implies that for cases with S = 2.5D, (and also S = 5D for the two staggered rows) neighbouring wakes start touching around this downstream location. This also drastically changes the wake spreading rate, as wakes essentially start to merge.

**Reviewer 2**

In the manuscript, the authors conduct a series of LES calculations to understand blockage in wind farms. The simulation approach uses a slip wall top boundary condition to mimic the capping inversion and varies the boundary layer height and spanwise turbine spacing to change the overall blockage. Both spanwise infinite, spanwise finite, and staggered configurations are considered. In addition, the authors develop an analytical model to quantify blockage affects on turbine/farm performance. Overall, the manuscript is technically sound, but there are several issues that must be addressed prior to publication.

**Reviewer Point P 2.1** — The governing equations are missing the viscous term. Is this neglected in the approach?

Reply: Yes, we clarified this in the updated manuscript:

Note that the direct effect of viscosity on resolved scales in the LES is negligible; all the dissipation is handled by the subgrid-scale stresses.

**Reviewer Point P 2.2** — The approach to provide an inflow profile from a precursor within a code with streamwise periodic boundary conditions seems needless complex. Why not simply use the finite difference method also in the streamwise direction with inflow/outflow boundary conditions? This almost certainly does not affect the simulation results, but it is cumbersome to explain.

**Reply**: We thank the reviewer for this remark. We clarify this in the updated manuscript:

To break the periodicity in the streamwise direction and prescribe an inflow condition, we use a fringeregion technique to drive the main domain by turbulent fully developed statistically steady flow fields obtained from a concurrent precursor simulation (Stevens et al., 2014). Using a precursor simulation ensures that turbulent inflow conditions are generated directly by the Navier-Stokes equations, thereby enhancing the accuracy and realism of the inflow turbulence (Munters et al., 2016).

**Reviewer Point P 2.3** — In the Smagorinsky model, how is the scalar grid spacing computed?

**Reply**: The scalar grid spacing is computed as follow:  $\Delta = (\Delta x \Delta y \Delta z)^{1/3}$ . We added this to the updated manuscript

**Reviewer Point P 2.4** — Figure 3 is quite misleading in how it is drawn and took me several attempts to interpret it. I interpreted the figure as the x-z plane, which leads to confusion with the text with regards to the area for the different configurations (e.g., A1=A2 for the spanwise infinite farm). The authors should add additional information to this figure to help with interpretability including both the x-z and x-y planes as well as figures specific to spanwise infinite and finite cases.

**Reply**: We added a x-z axis to Fig. 1 and a x-y axis to Fig. 3 (see Fig. RR2 and RR3). We also extended the caption of Fig. 3. Additionally, to clarify the control volume setup, an appendix with

Figure RR2: Sketch of the side view of a single turbine row (infinitely wide) inside an idealised ABL with rigid-lid top condition.

Figure RR3: Top view of the general setup of the control volume around a turbine to define blockage effects.

LES-based streamline plots for turbines in the infinite and finite row cases has been added to the revised manuscript.

**Reviewer Point P 2.5** — Analysis of the spanwise finite case is rather interesting but underdeveloped. All of the turbines have induction factors considerably larger than the corresponding infinite case. How can this be? This suggests a collective blockage effect of the entire farm. Will a spanwise finite farm ever look infinite due to the collective blockage effect? If a finite farm can look infinite, how large does it need to be? (The implications of this question are rather significant given the number of studies using spanwise periodic farms!)

**Reply**: We thank the reviewer for this comment. We have extended the analysis of the spanwise finite case in the updated manuscript:

As expected, the wakes of turbines at the edge of the row have more space for lateral expansion, and thus experience a lower  $|\Delta p_{NW}|$  than those at the centre, where expansion is constrained by neighbouring turbines and the boundary layer height. This explains the overall lower axial induction

found in the finite row cases compared to the equivalent infinite row cases, as seen in Fig. 5. All turbines in a finite row have less neighbouring turbines, resulting in a less constrained flow around the turbines. This can be considered as the collective blockage effect. Turbines within an infinite row are thus an extreme example of the centre turbines in a finite row. Additional simulations are needed to quantify the transition of a finite row into an infinite, but this is beyond the scope of the current study.

**Reviewer Point P 2.6** — The most significant technical deficiency is the lack of explanation for the overprediction of power and thrust by the analytical model. No commentary is provided to explain the overprediction, which is somewhat puzzling since some of the inputs came directly from the LES. This discrepancy must be explained.

**Reply**: We thank the reviewer for this remark. We have added an additional discussion section to the updated manuscript (Sect. 4.3), investigating this discrepancy, including a detailed momentum budget analysis.

**Reviewer Point P 2.7** — Finally, there are a few typographical errors, and some of the results are presented in mixed dimensional and non-dimensional form.

**Reply**: The typographical errors have been removed from the manuscript. The results in the graphs are now solely presented in non-dimensional form.

**Reviewer 3**

Overall, this is a useful and interesting study. I have comments and questions on the setup and assumptions made in the model development that should be further clarified. Further, given the poor agreement of the model in the finite farm case (Figure 11), even when provided the LES pressure data, it seems clear that the model is not adequate in this setting. It would be more helpful if the authors return to the fundamentals of the model derivation to reveal why the model is not accurate in this setting. This can guide improved modeling in this study and in future studies. In other words, the authors next goal is to couple the model from this paper with a separate pressure model for finite wind farms, but it appears that even with the right pressure, the model is not very accurate. More research should be performed in this paper to reveal the cause of the breakdown of the model.

I hope the authors can consider the following comments in a revision.

**Reviewer Point P 3.1** — Line 36: The sentence starting "Defining [...]" is very long and complex, consider re-phrasing for clarity.

Reply: This has been changed in the updated manuscript:

They defined the non-local efficiency  $\eta_{nl} = P_1/P_{\infty}$  as the ratio between the power of a free standing turbine  $P_{\infty}$  and the average turbine power of the first row of a wind farm  $P_1$ , and a wake efficiency  $\eta_w = P_{tot}/(N_tP_1)$  as the ratio between the average turbine power in the wind farm  $(P_{tot}/N_t)$  and the average turbine power of the first row. A strong negative correlation was found between the unfavourable upstream pressure gradient and  $\eta_{nl}$ , and a strong positive correlation between the favourable pressure gradient within the farm and  $\eta_w$ .

**Reviewer Point P 3.2** — Line 51: "Looking at a control volume [...]" I appreciate that the introduction utilizes fundamental arguments to justify the setup. This discussion can be improved, because it is not "directly clear" from the schematics that there should be an unfavorable pressure gradient. Rather, this is clear from an inspection of the conservation equations along with the geometry.

**Reply**: Indeed, this is a direct result from Newton's second law, the presence of the turbine thrust force, and continuity. We have changed the paragraph accordingly in the updated manuscript, see lines 51 - 65.

**Reviewer Point P 3.3** — Line 54: "so that outflowing momentum is larger than inflowing momentum"

Why is this guaranteed to be the case for all parameter values?

**Reply**: For the same mean velocity (continuity), the momentum scales with the square of the streamwise velocity, so that in the presence of a wake deficit there should be a higher outgoing momentum than the free stream.

**Reviewer Point P 3.4** — Eq. 1: The analysis, boundary conditions, and assumptions to arrive at Equation 1 are not clear

**Reply**: We have changed the paragraph for more clarity in the updated manuscript, see lines 51 - 65.

**Reviewer Point P 3.5** — Line 61: How does Equation 1 tell you that  $|p_{NW}| \gg |p_{FW}|$ ?

**Reply**: We thank the reviewer for this comment. It is indeed not possible to conclude  $|\Delta p_{NW}| \gg |\Delta p_{FW}|$  from the equations, only  $|\Delta p_{NW}| > |\Delta p_{FW}|$ . This was a typo. We clarify this derivation further in the updated paragraph, see lines 51 – 65.

**Reviewer Point P 3.6** — Line 80: How are the authors able to conclude from Figure 1 that the far wake pressure gradient will be much smaller than the near wake pressure gradient for all parameter values?

**Reply**: See previous point.

**Reviewer Point P 3.7** — Line 109: The Shapiro correction factor was derived based on Froude momentum theory without blockage, but the present simulations include blockage. This sensitivity should be expanded upon. Does the correction factor affect the results?

**Reply**: We thank the reviewer for this thoughtful comment. Indeed the Shapiro correction factor does affect the results, by introducing additional uncertainty for the highest blockage case. We extended the updated manuscript with a final discussion section (Sect. 4.3), where we analyse this further.

**Reviewer Point P 3.8** — Section 2.2: The authors have made the choice to fix the friction velocity in simulations with different domain heights, rather than fixing the pressure gradient. This choice could be justified in more detail.

**Reply**: We have added a clarification in the updated manuscript:

All precursor simulations are driven by a constant pressure gradient  $dp_{\infty}/dx_1 = -u_{\tau}^2/H$ , with a friction velocity  $u_{\tau} \approx 0.275 \text{ ms}^{-1}$  typical for offshore conditions (Lanzilao and Meyers, 2024). Fixing  $u_{\tau}$  yields approximately the same hub-height wind speed for a given  $z_0$  and varying H simulations.

**Reviewer Point P 3.9** — Figure 3: It is unclear to me why the  $p_{side}$  approximation is made in this 'general setup' figure. What is the justification for this approximation?

**Reply**: We have slightly reformulated the expression of  $p_{side}$  in the updated manuscript as  $p_{side} = p_{in} + \alpha \Delta p_{NW}$  with  $0 < \alpha < 1$ , since the pressure increases in the streamwise direction from  $p_{in}$  to  $p_{in} + \Delta p_{NW}$ . Indeed, in the model we use an approximation of  $p_{side}$  with  $\alpha = 0.5$ . The justification for the choice of  $\alpha$  is further explained in the new discussion section (Sect. 4.3):

... the model assumption of the side pressure  $p_{side} = p_{in} + \alpha \Delta p_{NW}$  with  $\alpha = 0.5$ , is not entirely correct. The LES shows that  $p_{side}$  shifts closer to  $p_{in}$ , i.e.  $\alpha < 0.5$ , however  $\alpha$  depends on the blockage strength. Nevertheless, the contribution of  $p_{side}$  in the streamwise momentum balance is significantly smaller than the other terms, due to the small streamwise projection of the side surfaces.

**Reviewer Point P 3.10** — Line 165: The geometry used in this modeling is non-standard and should be introduced more clearly. In classic axial momentum theory, the streamtube and control

volumes are typically cylindrical. Here, the authors set  $A_1 = S * H$  (rectangular). The implications of this choice are not described.

**Reply**: We have added a justification in the updated manuscript:

We choose the area  $A_1 = SH$ , which corresponds to the 'available' inflow for the turbine. Note that the outlet area  $A_2$  is a priori unknown, but expected to be larger than  $A_1$ . In the case of an infinite row of turbines,  $A_2 = A_1$  is obtained. Moreover, similarly to Garrett and Cummins (2007) and Werle (2010),  $A_1$  and  $A_2$  do not need to be cylindrical as the momentum theory is a one dimensional analysis which does not intrinsically define the cross-sectional shapes.

**Reviewer Point P 3.11** — Figure 3 / Line 165: More generally, the choice of the larger control volume following streamlines is non-standard, and requires the authors to make arguments about the side pressure. I'm also having trouble understanding what it means for  $A_1$  to be the "the 'available' inflow for the turbine," equal to spacing S times domain height H, but then for the control volume to expand into  $A_2$  ( $A_2 > A_1$ ). Thus, for the finite farm, the streamtube for one wind turbine will overlap with the streamtube from another turbine. But streamlines cannot cross each other.

**Reply**: We thank the reviewer for this remark. To clarify the control volume setup, an appendix with LES-based streamline plots for turbines in the infinite and finite row cases has been added to the revised manuscript.

**Reviewer Point P 3.12** — Line 180: typographical error

Reply: This has been changed in the manuscript.

**Reviewer Point P 3.13** — Line 183: The approximation of  $p_{side}$  requires much more justification based on theory and the LES data

**Reply**: See previous point P3.9

**Reviewer Point P 3.14** — Line 223: Since  $U_d$  is rotor averaged, it seems logical that  $U_{in}$  should be the rotor average of the freestream wind from the precursor

**Reply**: We have now clarified this decision in the updated manuscript, on line 311:

We also briefly compared with calculating  $C_P$  and  $C_T$  in the LES using a rotor-disk-averaged velocity at rotor height in the precursor domain. In the results (not shown here) this produced similar  $C_P$  and  $C_T$ , with less than a 1% difference for the highest blockage case.

**Reviewer Point P 3.15** — Line 230: In what spatial position is the maximum velocity upstream of the turbine occurring?

**Reply**: We updated the sentence to clarify this:

Therefore, for the second row,  $U_{in}$  is defined as the maximum velocity upstream of the turbine, which is located at the end of the near wake of the turbines in the first row (see Fig. 4d).

**Reviewer Point P 3.16** — Figure 5: As noted in the discussion, the finite and infinite farms differ in their induction factors, which is not explained by the  $A/A_d$  parameter, which only describes blockage on the turbine level. It seems natural to define a farm geometric blockage parameter that is the total cross section of the domain (Ly \* H) divided by the total surface area of all turbines  $(N * \pi * D^2/4 \text{ with N turbines}).$

**Reply**: We believe that a farm geometric blockage parameter defined a as  $(Ly * H)/(N * \pi * D^2/4)$  is not meaningful to characterize blockage in the finite row cases with infinitely wide domains. We have added a clarification in the updated manuscript:

We note that an alternative geometric blockage parameter based on the cross-section of the domain, i.e.,  $A_{domain}/A_{farm} = 4L_yH/N\pi D^2$ , with  $L_y$  the domain width, as, e.g. used for analysis of tidal channels (see, e.g., Nishino and Willden (2012)) is not meaningful here, since for our finite wind-farm simulations we are considering  $L_y \rightarrow \infty$ .

**Reviewer Point P 3.17** — Section 4.1.2: "Here we observe that a classical Gaussian shape function provides good fits along the downstream direction." The fit should be quantitatively evaluated and those quantities should be reported, as in Brugger et al. (2019); Dar and Porté-Agel (2024).

**Reply**: We have added the coefficient of determination  $(R^2)$  of the gaussian fit on the LES data, in Fig. 6.

**Reviewer Point P 3.18** — Line 271: "As H increases, C(x) remains approximately the same, while  $\delta_y(x)$  spreads faster." This is an interesting/unexpected result, as we typically think of the wake recovery rate and the wake spreading rate as connected quantities (more comments on this later in regards to interpretation of the results/conclusions). Perhaps this is because the authors only evaluate horizontal wake width, rather than the three-dimensional wake shape?

**Reply**: We thank the reviewer for this observant remark. This is indeed an interesting result, which requires more clarity. We have extended this paragraph in the updated manuscript:

As H increases, C(x) shows a slightly steeper decline, while  $\delta_y(x)$  spreads faster. In particular, the difference between H = 350 m and the other two BL heights is clearly visible. This can be understood by looking at the evolution of the vertical velocity profiles (in the wake centre) shown in Fig.8 for infinite row cases with S = 5D. In particular for cases H350 is observed that vertical spreading of the wake is limited by the presence of the rigid-lid, while the maximum wake deficit seems minimally impacted. Note that the turbine tip height (270 m) is not much lower than the BL height in this case. Thus for this case, the turbine far wake behaves less as an axisymmetric and more as a planar wake, which is known to have a lower spreading rate (Pope, 2000). We remark that the rigid-lid only reduces the overall wake spreading at low H, while the wake recovery seems only slightly affected. We suggest that this imbalance is due to the unusual shape of the wake in the vertical direction for H350.

**Reviewer Point P 3.19** — Line 274: "In particular for cases H350 is observed that vertical spreading of the wake is limited by the presence of the rigid-lid."

This especially motivates the need to evaluate the reliability of the Gaussian fitting, as in comment 17.

**Reply**: We have added the coefficient of determination  $(R^2)$  of the gaussian fit on the LES data, in Fig. 6.

**Reviewer Point P 3.20** — Line 315: The pressure is calculated at x = 4D (what the authors say is the end of the near wake). But does the near wake length depend on the blockage?

**Reply**: We thank the reviewer for this comment. Yes the near wake length depends on the blockage. This was poorly phrased in the manuscript. We have rephrased this section in the updated text:

To appraise this large scale pressure system, we evaluate for the finite row cases in Fig. 10a the pressure drop  $-\Delta p_{nw}(y)$ , only containing the pressure perturbations superimposed on the background pressure gradient (see Eq. 4).  $-\Delta p_{nw}(y)$  is calculated as the difference between the domain inlet and the end of the near wake (defined as the location of maximum wake deficit), averaged over H, and shown as a function of the spanwise direction y, where y = 0 is located at the row centre.

**Reviewer Point P 3.21** — Figure 10 and associated discussion: this result suggests that the radial normal Reynolds stress is relevant to the analysis. But these Reynolds stresses were neglected in the model. Discussion of this would be useful.

**Reply**: We have added an additional discussion section (Sect. 4.3) to the updated manuscript. There we perform a detailed momentum budget analysis. In brief, the radial normal Reynolds stresses do not play a role in the axial momentum balance.

**Reviewer Point P 3.22** — Figure 11, where the model is closed with LES pressure, implies that the model is not adequate for the finite farm (i.e. an assumption/approximation ma